# Early Endosomal Vps34-Derived Phosphatidylinositol-3-Phosphate Is Indispensable for the Biogenesis of the Endosomal Recycling Compartment

**DOI:** 10.3390/cells11060962

**Published:** 2022-03-11

**Authors:** Marina Marcelić, Hana Mahmutefendić Lučin, Antonija Jurak Begonja, Gordana Blagojević Zagorac, Pero Lučin

**Affiliations:** 1Department of Physiology and Immunology, Faculty of Medicine, University of Rijeka, 51000 Rijeka, Croatia; mmarcelic@uniri.hr (M.M.); hana.mahmutefendic@uniri.hr (H.M.L.); gordana.blagojevic@uniri.hr (G.B.Z.); 2Department of Physiology, Immunology and Pathophysiology, Faculty of Medicine, University Center Varaždin, University North, Jurja Križanića 31b, 42000 Koprivnica, Croatia; 3Department of Biotechnology, University of Rijeka, Radmile Matejčić 2, 51000 Rijeka, Croatia; ajbegonja@bioteh.uniri.hr

**Keywords:** phosphatidylinositol-3-phosphate, Vps34, VPS34-IN1, Rab11a endosomes, endosomal recycling compartment

## Abstract

Phosphatidylinositol-3-phosphate (PI3P), a major identity tag of early endosomes (EEs), provides a platform for the recruitment of numerous cellular proteins containing an FYVE or PX domain that is required for PI3P-dependent maturation of EEs. Most of the PI3P in EEs is generated by the activity of Vps34, a catalytic component of class III phosphatidylinositol-3-phosphate kinase (PI3Ks) complex. In this study, we analyzed the role of Vps34-derived PI3P in the EE recycling circuit of unperturbed cells using VPS34-IN1 (IN1), a highly specific inhibitor of Vps34. IN1-mediated PI3P depletion resulted in the rapid dissociation of recombinant FYVE- and PX-containing PI3P-binding modules and endogenous PI3P-binding proteins, including EEA1 and EE sorting nexins. IN1 treatment triggered the rapid restructuring of EEs into a PI3P-independent functional configuration, and after IN1 washout, EEs were rapidly restored to a PI3P-dependent functional configuration. Analysis of the PI3P-independent configuration showed that the Vps34-derived PI3P is not essential for the pre-EE-associated functions and the fast recycling loop of the EE recycling circuit but contributes to EE maturation toward the degradation circuit, as previously shown in Vps34 knockout and knockdown studies. However, our study shows that Vps34-derived PI3P is also essential for the establishment of the Rab11a-dependent pathway, including recycling cargo sorting in this pathway and membrane flux from EEs to the pericentriolar endosomal recycling compartment (ERC). Rab11a endosomes of PI3P-depleted cells expanded and vacuolized outside the pericentriolar area without the acquisition of internalized transferrin (Tf). These endosomes had high levels of FIP5 and low levels of FIP3, suggesting that their maturation was arrested before the acquisition of FIP3. Consequently, Tf-loaded-, Rab11a/FIP5-, and Rab8a-positive endosomes disappeared from the pericentriolar area, implying that PI3P-associated functions are essential for ERC biogenesis. ERC loss was rapidly reversed after IN1 washout, which coincided with the restoration of FIP3 recruitment to Rab11a-positive endosomes and their dynein-dependent migration to the cell center. Thus, our study shows that Vps34-derived PI3P is indispensable in the recycling circuit to maintain the slow recycling pathway and biogenesis of the ERC.

## 1. Introduction

The cellular membrane system is highly dynamic, and the continuous flux of membranes between intracellular compartments and plasma membranes (PM) maintains a homeostatic balance of membrane components according to cellular needs. The endocytic activity of PM internalizes, and the recycling process within the internal compartments returns the PM equivalent several times within an hour [1]. Both endocytic uptake and membrane flux through internal compartments involve the continuous changes in membrane domain identity by adjusting lipid composition and membrane recruitment of cytoplasmic proteins. These processes are controlled by the cascade-like recruitment of small GTPases from the Rab and Arf family and their regulatory guanine-nucleotide exchange factors (GEFs) and GTPase-activating proteins (GAPs) [2]. The spatiotemporal activation and inactivation of these proteins are associated with the temporal recruitment of effector proteins that affect membrane functions, including enzymes that regulate lipid components within membranes. Among lipids, phosphorylation of phosphatidylinositol (PI) at various positions appears to be essential for the establishment of the identity tag at membrane domains [3,4].

Endocytosed membranes are first collected in peripheral membranous stations, known as pre-early endosomes (pre-EEs) [5], and differentiate into early endosomes (EEs) by acquiring Rab5a and the subsequent recruitment of Vps34. Vps34 forms a complex of class III phosphoinositide 3-kinase (PI3K) and phosphorylates PI at the 3′ position to generate phosphatidylinositol-3-phosphate (PI3P) at membranes [4,6]. PI3P is a hallmark of further EE membranous stations. It provides a platform for the recruitment of proteins with specific PI3P-binding domains, such as the FYVE (Fab1, YOTB, Vac1, and EEA1) zinc-finger domain and the PX (Phox homology) domain [7]. These proteins drive further maturation steps of EEs, including homotypic fusion, cargo sorting, plus-end-directed movements along the microtubule, tubulation, and the development of outgoing endosomal carriers in various directions [6].

The maturation of EEs involves a dynamic reconfiguration of parts of their membranes, known as membrane domains, through changes in membrane heterogeneity, the recruitment of small GTPases from the Rab and Arf subfamilies, and the assembly of tethering and cargo-sorting protein complexes [2,8]. The biochemical sequences of this conversion process are still not fully understood. For example, Rab5 is replaced by Rab4 or Rab11 [2] to generate tubular recycling endosomes, or by Rab8 [9], Rab35 [10], Rab10 [11], Rab14 [12], and Rab15 [13] to generate a complex cluster of membranes in the cell center, known as the endosomal recycling compartment (ERC). The replacement of Rab5 with Rab7 or Rab9 drives the conversion of EEs to late endosomes (LEs) [2]. All of these conversion reactions are associated with the loss of Vps34, a decrease in PI3P production, and the recruitment of a PI-phosphatase that dephosphorylates PI3P, followed by further phospholipid modifications required for endosome maturation [7,14].

Although PI3P is a recognized feature of EEs, its role in the final maturation of EEs remains insufficiently understood. The fundamental role of PI3P in the physiology of EEs was established in studies using the pharmacological PI3K inhibitors wortmannin and LY294002. However, these inhibitors were not selective for class III PI3K and not even for PI3Ks [15]. Therefore, understanding the EE physiological pathways based on these inhibitors is blurred by their numerous undefined off-target effects. The knockdown [16,17] and knockout [18,19] of Vps34 expression provided more specific insights into the role of PI3P in EE physiology. However, these approaches were associated with long-term perturbations of the endosomal system and led to the conclusion that PI3P at EEs is essential for LE biogenesis, while other PI3P functions within EEs may be compensated. Inhibition of Vps34 by microinjected inhibitory antibodies [20] and acute activation of inositol-3-phosphatase myotubularin 1 (MTM1) at EEs [21] provided short-term tools to evaluate the physiological role of PI3P at EEs. Finally, the introduction of rapidly acting and highly specific Vps34 inhibitors [22,23,24] opened the door for studying EE physiology under the short-term disruption of the endosomal system. The initial studies of rapid PI3P depletion by PI3K inhibitors and the follow-up studies that have since become available [25,26,27] were mainly limited to the effects of PI3P depletion on LE-associated functions, whereas the impact on EE functions remains understudied.

In this study, the EE recycling circuit was analyzed under conditions of rapid PI3P depletion by the class III PI3K (Vps34) inhibitor VPS34-IN1. This inhibitor targets the catalytic activity of a Vps34 and depletes the Vps34-derived PI3P pool within 1 min [22]. It does not affect its protein scaffolding function and has no significant effect on other 25 lipid kinases and more than 340 protein kinases [22]. Our analysis of the endosomal recycling circuit after the short-term depletion of PI3P reveals the rapid reorganization of Rab11a-dependent endosomal pathways, including cargo sorting and biogenesis of the endosomal recycling compartment.

## 2. Materials and Methods

### 2.1. Cells

Balb 3T3 fibroblasts and HeLa cells, obtained from the American Type Culture Collection (ATCC, Manassas, VA, USA), were grown in DMEM supplemented with 10% (*v/v*) fetal bovine serum (FBS), 2 mM l-glutamine, 100 mg/mL streptomycin, and 100 U/mL penicillin (all reagents from Gibco/Invitrogen, Grand Island, NY, USA).

### 2.2. Antibodies and Reagents

Antibody reagents for the membranous organelle markers were purchased from various vendors. The sources of the primary antibody reagents and the references for their validation are listed in Appendix A. 

VPS34-IN1 was purchased from Cayman Chemical (Tallinn, Estonia) and dissolved in DMSO as a 10 mM stock solution. Control cells were treated with an appropriate dilution of DMSO in the tissue-culture medium.

Alexa Fluor (AF)^488^-, AF^594^-, and AF^555^-holo-transferrin (hTf) and unlabeled holo-transferrin (hTf) were purchased from Molecular Probes (Eugene, OR, USA). AF^488^-, AF^594^-, and AF^555^-conjugated secondary antibody reagents against mouse IgG_2a_, mouse IgG_2b_, mouse IgG_1_, rat IgG, rabbit IgG, and chicken IgG were obtained from Molecular Probes (Leiden, NL, USA), and AF^680^-conjugated IgG_1_ and IgG_2a_ were obtained from Jacksons Laboratory (Bar Harbor, ME, USA). Ciliobrevin D was purchased from Merck Millipore (Burlington, MA, USA) and dissolved in DMSO as 10 mM stock. Cytofix/cytoperm were purchased from Becton Dickinson & Co. (Franklin Lakes, NJ, USA). Propidium iodide and other chemicals were from Sigma-Aldrich Chemie GmbH (Taufkirchen, Germany).

### 2.3. Transfection of 2xFYVE and p40PX PI3P-Binding Domains 

Murine stem cell retroviral vectors (MSCVs) containing EGFP, YFP, the EGFP- tagged 2xFYVE domain of Hrs (EGFP-2xFYVE), and the EGFP-tagged 2xFYVE domain of Hrs with a double mutation [EGFP-2xFYVE(C215S)], the YFP-tagged PX domain of p40phox (YFP-PX), and the YFP-tagged PX domain of p40phox with a mutation (YFP-PX (R57Q) were previously reported in [28]. The indicated constructs were used for the transient transfection of Balb3T3 fibroblast cells with Lipofectamine 2000 transfection reagent (Invitrogen) according to the manufacturer’s instructions.

### 2.4. Labeling of Transferrin-Loaded Compartments and Internalization of Transferrin Receptors

For labeling Tf-loaded endosomal compartments, cells were incubated for 45 min with 50 μg/mL of either hTf-AF^594^, hTf-AF^555^, or hTf-AF^488^. Internalization of cell surface transferrin receptors (TfRs) was monitored by either ligand- or mAb-labeling assays as previously described [29,30]. Briefly, cells were incubated with anti-TfR mAbs (2 μg/mL) at 4 °C for 30 min, unbound mAbs were removed by washing three times with the cell culture medium, and internalization was initiated by incubation at 37 °C. When internalization of cell surface TfRs was monitored by fluorescently labeled holo-transferrin (hTf), cells were incubated for 30 min at 4 °C with saturating concentrations of either hTf-AF^594^, hTf-AF^555^, or hTf-AF^488^ (50 μg/mL) diluted in FCS-free tissue culture medium. After labeling at 4 °C, the cells were washed three times with a cold medium and incubated at 37 °C-warm medium containing 10% FBS (chase).

### 2.5. Flow Cytometric Quantification of Recycling

Quantitative analysis of recycling was performed on de-adhered cells using procedures that minimally alter plasma membrane (PM) function and dynamics. Cells were washed with PBS containing 5 mM EDTA and detached by treatment with trypsin-EDTA solution (0.137 M NaCl, 0.003 M KCl, 1.5 mM KH_2_PO_4_, 3.2 mM Na_2_HPO_4_, 0.125% porcine trypsin, 3.4 mM Na-EDTA, 0.045 mM phenol red) at 37 °C for 1–2 min. To minimize the time required for temperature shifts, cells were always resuspended in pre-warmed or pre-cooled media in pre-warmed/cooled tubes.

Recycling of TfR was quantified by detecting the loss of fluorochrome-conjugated Tf from cells after pulse internalization [29]. Cells were incubated with hTf-AF^488^ or hTf-AF^555^ (50 μg/mL) at 37 °C for 45 min to load the intracellular compartments and then washed three times in a medium containing unlabeled hTf. The amount of internalized hTf was quantified by flow cytometry (DMFI_int, t = 45_). Loss of fluorescence due to recycling was determined after incubation at 37 °C for various periods (DMFI_rec, t = x_) in the medium containing 200 μg/mL unlabeled hTf. The percentage of recycled proteins was calculated as (1-DMFI_rec, t = x_/DMFI_int, t = 45_) × 100. The amount of recycled proteins R(t_n_) was calculated as R(t_n_) = 100 − 100(1 − *k*_r_Δt).

### 2.6. Immunofluorescence and Confocal Analysis

Cells grown on coverslips were fixed with 4% formaldehyde (20 min at r.t.) and permeabilized with 0.5–1% Tween 20 at 37 °C for 20 min. After permeabilization, cells were incubated with primary Ab reagents for 60 min. Unbound Ab reagents were washed with PBS, and cells were incubated with an appropriate fluorochrome-conjugated secondary reagent for 60 min. After washing three times in PBS, cells were embedded in Mowiol (Fluka Chemicals, Selzee, Germany)-DABCO (Sigma Chemical Co., Steinheim, Germany) in PBS containing 50% glycerol and analyzed by confocal microscopy. 

Images were acquired using an Olympus Fluoview FV300 confocal microscope (Olympus Optical Co., Tokyo, Japan) equipped with Ar 488, He/Ne 543, and He/Ne 633 lasers, or on a Leica DMI8 inverted confocal microscope (confocal part: TCS SP8; Leica Microsystems GmbH, Wetzlar, Germany) equipped with a UV laser (diode 405), Ar 488, DPSS 561, and He/Ne 633 lasers. Images were acquired using Fluoview software, version 4.3 FV 300 (Olympus Optical Co., Tokyo, Japan), PLAPO60xO objective, appropriate filters, and PMT detectors. The confocal aperture was set to 2. Otherwise, images were acquired using Leica Application Suite X (LAS X) software (Leica Microsystems GmbH, Wetzlar, Germany; https://www.leica-microsystems.com/products/microscope-software/p/leica-las-x-ls/; accessed on 10 March 2020), the HC PLAPO CS2 (63×/1.40 oil) objective, and 4 detectors (2× PMT and 2× HyD). The z-series of 0.3–0.5 μm optical sections were acquired sequentially with an offset below 5% and medium scan speed (1.65s/scan). The images (512 × 512 pixels) were acquired at different zoom values (zoom factor: 0.75–6.0) with pixel sizes ranging from 481.47 nm × 481.47 nm to 60.18 nm × 60.18 nm. 

### 2.7. Image Analysis

Images were exported as TIFF and analyzed using Fiji (ImageJ2) and available plugins (https://imagej.net/software/fiji/; accessed on 10 March 2022) without any image rendering or additional processing. Fluorescence images in the focal plane were used to display the images and colocalization. 

Colocalization events were quantitatively evaluated on images with a pixel size of 120.37 × 120.37 nm, using a global statistical approach that performs intensity correlation coefficient (ICCB)-based analysis. We used the JACoP plugin (http://rsb.info.nih.gov/ij/plugins/track/jacop.html; accessed on 10 March 2022) [31] to calculate the Manders overlap coefficients (M1 and M2) within the entire z-stack for three-dimensional (3D) analysis of colocalization. The background was partially eliminated during image acquisition by adjusting detector settings to detect maximum fluorescence intensity in the red and green channels. The best-fit lower threshold to eliminate most of the signal background (Costes automatic thresholding method) was determined using the thresholding tool and confirmed by visual inspection.

Quantification of fluorescence intensity was performed on images with a pixel size of 120.37 × 120.37 nm using the Auto Threshold tool and Measure plugins according to published protocols [32]. Briefly, the cells of interest were selected using the freeform selection tool, and the area, integrated density, and mean gray value were measured for each selected cell. For background correction, the area adjacent to the selected cells that had no fluorescence was used. Total corrected cell fluorescence (TCCF) for each cell of interest was calculated using the following formula: integrated density—(area of selected cell x mean fluorescence of background values).

Quantification of the juxtanuclear area by concentric circles was performed on freehand selected regions on images with a pixel size of 120.37 × 120.37 nm using the Otsu Auto Threshold and Measure plugins of ImageJ software. Concentric circles (0–54, 55–108, and 109–162 pixel distance) were centered on the area with the highest fluorescence signal. The intensity within the circles was quantified relative to the total intensity in the whole cells.

### 2.8. Western Blot

Cell extracts for WB analysis were prepared in RIPA lysis buffer supplemented with protease and phosphatase inhibitors, separated by SDS-PAGE, and blotted at 60 to 70 V for 1 h onto a polyvinylidene difluoride (PVDF-P) WB membrane (Merck Millipore). Membranes were incubated with 1% blocking reagent (Roche Diagnostics GmbH, Mannheim, Germany) for 1 h, followed by a 1 h to overnight incubation with primary Abs, three wash cycles (TBS with 0.05% Tween 20 [TBS-T buffer]), and a 45 min incubation with peroxidase-conjugated secondary reagent diluted in TBS buffer with 0.5% blocking reagent. After washing three times with TBS-T buffer (pH 7.5), membranes were incubated with SignalFire Elite ECL reagent (Cell Signaling Technology, Danvers, MA, USA) for 1 min and enveloped in plastic wrap. Signals were detected using the Transilluminator Alliance 4.7 (Uvitec Ltd., Cambridge, UK).

### 2.9. Data Presentation and Statistics

Data are presented as mean ± standard error of the mean (SEM). Data comparison was performed using a two-tailed Student’s *t*-test when two samples were compared and one-way ANOVA analysis of variance for data with more than two experimental groups. Differences were considered significant when *p*-values were <0.05 (* *p* ≤ 0.05; ** *p* < 0.01; *** *p* < 0.001).

## 3. Results

### 3.1. Pharmacological Inhibition of Vps34 Rapidly and Reversibly Depletes Endosomal PI3P Pool and Alters PI3P-Associated Functions

First, we examined whether depletion of the Vps34-derived PI3P pool affects the EE association of PI3P-dependent proteins and leads to functional changes in a PI3P-dependent pathway. We treated YFP-PX_P40phox_ (p40PX) or EGFP-2xFYVE_Hrs_ (2xFYVE) transfected cells with 10 μM Vps34-IN1 (IN1) and monitored the endosomal association of PI3P-binding probes by immunofluorescence. To confirm the effect of IN1, cells were also treated with another Vps34 inhibitor, SAR405. As controls, we used YFP-PX_P40phox_^R57Q^ (p40PX^R57Q^) and EGFP-2xFYVE_Hrs_^C215S^ (2xFYVE^C215S^), the same constructs with a mutation in the PI3P-binding sites [33,34], as well as vectors expressing EGFP or YFP alone. Four hours after the addition of the inhibitors, both p40PX (Figure 1A) and 2xFYVE dispersed in the cytosol (Figure 1B), as when YFP or EGFP were expressed alone or in association with the mutant modules p40PX^R57Q^ or 2xFYVE^C215S^ (Appendix A). Since prolonged treatment with IN1 did not affect the viability of Balb 3T3 cells (data not shown), we further examined the use of this inhibitor for the short- and long-term depletion of PI3P. The effect of 10 μM IN1 was rapid and dispersed PI3P-binding modules already after 10 min treatment (data not shown), consistent with previous observations [22]. A similar effect was observed after 3 μM treatment for 60 min, with a small fraction of p40PX and 2xFYVE associated with some cytoplasmic membranous structures (Appendix A). The remnants were significantly reduced after 4 h of treatment and largely disappeared after 5 and 10 μM treatments. Extensive vacuolization was observed after one hour in some of the cells and after four hours in most cells. These experiments demonstrate that pharmacological inhibition of class III PI3K by IN1 rapidly depletes PI3P from endomembranes and causes dissociation of FYVE- and PX-domain-containing proteins.

Cytoplasmic scattering of fluorescent PI3P-binding modules prevented the accurate estimation of PI3P depletion. To gain better insight, we performed a detailed analysis by monitoring the non-fluorescent endogenously expressed and well-defined protein EEA1, which contains the FYVE domain and recruits to PI3P at EEs [35,36]. Short-term treatment for one hour with 1 μM IN1 released ~89% of EEA1 from membranes, and cytoplasmic EEA1 was washed out by permeabilization, resulting in a marked decrease in cell-associated fluorescence (Figure 1C). A similar and even stronger effect was obtained by increasing the concentration of the inhibitor and the incubation time (Figure 1C). Nevertheless, at least 6% of the fluorescence remained associated with the cells (Figure 1C), and part of it was associated with membranous structures located mainly near the nucleus (Appendix A). This effect lasted for a long time and could be detected after 4 (Figure 1C) and 12 h (data not shown). 

Since the effect of treatment with 1 µM IN1 varied as a function of cell density, we used a concentration of 3 µM in all experiments. At this concentration, the EEA1 fluorescence signal rapidly decreased to ~44% of the control level at 5 min, to ~22% at 30 min, to ~12% at 60 min, and to 7% at 120 min (Figure 1D). Total corrected fluorescence (TCCF) was measured by subtracting the background of the same area outside the cell. Therefore, we also measured the fluorescence intensity of the cell area stained with the same secondary reagent. As expected, the background intensity was higher in the cell area (Figure 1D, BL). On this basis, we estimate that very little EEA1 was associated with the membranous elements of PI3P-depleted cells after 120 min of IN1 treatment. These data support the existence of a large pool of endogenous PI3P-associated protein that rapidly dissociates and a pool that gradually detaches from EE membranes after PI3P deprivation. After IN1 washout, EEA1 was detectable in some cells after 2, in almost all cells after 5, and restored to the control levels after 30 min (Figure 1E and Appendix A). Overall, the analysis of EEA1 demonstrates the rapid dissociation of endogenous PI3P-binding protein from EEs after the pharmacological inhibition of Vps34 and rapid recovery of PI3P production after IN1 washout. Western blot analysis showed that the displaced EEA1 was not degraded, even after 12 h of IN1 treatment (Figure 1F and Appendix A).

We also tested the effect of IN1 on the expression of Hrs, another FYVE domain-containing protein that is initially recruited to PI3P but localizes to EEA1-negative PI3P-independent microdomains of EEs [37]. In contrast to EEA1, Hrs remained associated with membranes in IN1-treated cells (Figure 1G). Considering that EEA1 localizes to dynamic PI3P-rich microdomains, whereas Hrs is rapidly displaced from these PI3P-rich domains into regions of restricted entry and exit after recruitment [38], these data suggest that the depletion of the Vps34-derived PI3P pool affects the recruitment of endogenous FYVE domain-containing EE proteins localized to dynamic PI3P-rich EE microdomains.

To investigate whether PI3P depletion also affects the membrane association of endogenous PX domain-containing proteins, we tested the effect of IN1 on sorting nexins (SNXs) acting in the recycling circuit. SNX1 and SNX3, two SNXs associated with dynamic exit membranes of EEs [39], are highly associated with Tf-loaded endosomes of control cells (Appendix A). In addition, we tested SNX4 with two different antibodies, but neither showed sufficient signal (data not shown), suggesting that SNX4-associated activities are not well expressed in Balb 3T3 cells. After 2 h of IN1 treatment, SNX1 and SNX3 almost completely detached from the membrane structures (Figure 1H and Appendix A), indicating that IN1 treatment rapidly depletes two SNXs essential for cargo recycling from EEs [40,41,42,43].

Altogether, our data confirm that IN1 efficiently depletes the Vps34-dependent PI3P pool of EEs. The depletion is rapid and quickly reversible, affecting PI3P-dependent effector proteins associated with the incoming and outgoing dynamics of EEs.

### 3.2. Depletion of Vps34-Derived PI3P Arrests Internalized Tf in Perinuclear Endosomes

To examine the effects of short-term PI3P depletion on the EE recycling circuit, we analyzed the best-established marker for this pathway, the transferrin receptor (TfR) [44]. In Balb 3T3 cells, fluorescent Tf was rapidly taken up by endocytosis and loaded into EEs within 2–3 min, into perinuclear vesicular and tubular EEs within 3–5 min, and into the juxtanuclear ERC after 8–10 min of exposure coincident with Rab11a [45]. During the 45 min exposure, approximately half of the internalized Tf is retained in the juxtanuclear EEs and ERC (Figure 2A, Figure 3 and Appendix A). Some internalized Tf was also located in peripheral endosomes, including pre-EEs, only a few of which could be visualized (Figure 2A, indicated by arrows).

Short-term treatment with 3 μM IN1 (60 min) had no effect on Tf uptake, as confirmed by quantitative analysis of confocal images (Figure 2B), indicating that PI3P depletion does not affect endocytosis and recycling processes upstream of EEs, i.e., from APPL1-positive very early endosomes [46,47]. However, in IN1-treated cells, internalized Tf-AF^594^ did not concentrate in the juxtanuclear area (Figure 2A). Still, it remained in scattered perinuclear endosomes, which were vacuolized in some cells after 60 min of treatment and in most cells after longer treatment (Appendix A). In some cells, enlarged Tf-loaded endosomes accumulated below the nucleus (Figure 2A, indicated by arrowheads). The concentric circle analysis [48,49] showed the redistribution of Tf-loaded endosomes from the juxtanuclear region to the broader perinuclear area (Figure 2C). This analysis suggests that the depletion of Vps34-derived PI3P does not affect Tf uptake and transport along the EE pathway, including homotypic fusion of Tf-loaded endosomes, but likely alters endosome maturation and positioning.

### 3.3. PI3P Depletion Does Not Inhibit Tf Recycling

To identify a change in EE maturation steps in PI3P-depleted cells, we first analyzed the recycling kinetics of internalized fluorescent Tf. After 45 min pulse labeling with Tf-AF^488^, cells were chased in the presence of unlabeled Tf and analyzed by flow cytometric assay [29]. As previously determined by immunofluorescence analysis (Figure 2B), flow cytometric quantification of Tf-AF^488^ labeling revealed equal amounts of cell-associated fluorescence in control and PI3P-depleted cells (Figure 2D), confirming our conclusion that PI3P depletion does not affect Tf endocytosis. After the 30 min chase, Tf-AF^488^ was lost from control and IN1-treated cells with indistinguishable kinetics (Figure 2D), suggesting that TfR recycling occurs from Tf-loaded endosomes of PI3P-depleted cells.

### 3.4. PI3P Depletion Traps Internalized Tf in Rab5a/Rab4-Positive Endosomes and Prevents the Loading of Rab11a-Positive Endosomes

To characterize the enlarged Tf-loaded endosomes of PI3P-depleted cells, we performed double immunofluorescence staining and 3D colocalization analysis of internalized Tf with major organelle markers that distinguish pre-EEs (APPL1), EEs (Rab5 and Rab4), ERC (Rab11a), and LEs (Rab7a and Lamp1). 

In control cells, a subset of pre-EEs recruited APPL1 at the cell periphery. Some of these structures were loaded with internalized Tf-AF^594^ (Figure 3A). However, the vast majority of internalized Tf was not colocalized with APPL1 (Figure 3E), suggesting Tf retention in downstream endosomes. In contrast, short-term (Figure 3A) and long-term (Appendix A) PI3P depletion by IN1 resulted in increased recruitment of APPL1 to a subset of enlarged perinuclear Tf-loaded endosomes, leading to high colocalization of APPL1 with internalized Tf (Figure 3E and Appendix A, respectively). The median total fluorescence signal of APPL1 increased 2.1-fold after 60 min of IN1 treatment and 2.36-fold after 24 h of IN1 treatment (Appendix A), which is consistent with the previously published observation of the back-conversion of EEA1 to APPL1 endosomes after PI3P depletion [21] and confirms the efficiency PI3P depletion by IN1. Since APPL1 does not bind to PI3P but requires Rab5 for membrane association [21], these data suggest that internalized Tf accumulates in a portion of back-converted Rab5-positive endosomes.

As expected, internalized Tf was highly localized in Rab5a-positive endosomes in the perinuclear area of control cells and PI3P-depleted cells (Figure 3B), with a similar degree of overlap (Figure 3E). In PI3P-depleted cells, these endosomes were enlarged, vacuolized, and often arranged a ring-like pattern around the nucleus, especially after long-term (24 h) PI3P depletion (Appendix A). Enlarged Tf-loaded endosomes were largely positive for Rab4 (Figure 3C,E), indicating the activity of Rab4-dependent recycling processes at these membranes. A substantial proportion of the enlarged perinuclear Rab5a-positive endosomes of PI3P-depleted cells were positive for APPL1 (Figure 3F) and negative for EEA1 (data not shown), resulting in the increased colocalization of APPL1 and Rab5a (Figure 3J). Some internalized Tf was also found in the punctate and reticular structures outside the Rab5a- and Rab4-positive endosomes (Figure 3B,C).

After 45 min of continuous uptake in control cells, approximately 40% of the internalized Tf accumulated in Rab11a-positive endosomes (Figure 3E), which were mainly located around the cell center (Figure 3D). However, very little internalized Tf colocalized with Rab11a in the short- (Figure 3E) and the long-term (Appendix A) PI3P-depleted cells. Rab11a-positive endosomes were enlarged, vacuolized, and displaced from the cell center around the nucleus into a ring-like pattern (Figure 3D, Appendix A). These endosomes were not accessed by the internalized Tf (Figure 3D, Appendix A) and appeared to be completely separated from the Rab5a endosomes (Figure 3G), as indicated by the decrease in colocalization between Rab5a and Rab11a (Figure 3J). These data indicate that most of internalized Tf is retained in Rab5a/Rab4 endosomes of PI3P-depleted cells and, together with the recycling data (Figure 2D), suggest that Tf exits into the recycling circuit without entering the Rab11a-dependent recycling pathway.

Internalized Tf did not colocalize with Rab7a (Figure 3E and Appendix A), Lamp1, and GM1 (Appendix A), suggesting that internalized Tf was not transported to LEs in PI3P-depleted cells. Colocalization analysis of Rab5a and Rab7a showed a relatively strong overlap between Rab5a and Rab7a in the perinuclear area of control cells (Figure 3H,J), which decreased in PI3P-depleted cells (Figure 3H,J), suggesting that internalized Tf is retained in enlarged Rab5a-positive endosomes with reduced recruitment of Rab7a. In contrast, the overlap of Rab5a with Rab9a, a small GTPase defining a subset of LEs distinct from the Rab7 subset [50], increased markedly after PI3P depletion (Figure 3J), suggesting that Rab9a is retained at the subdomain of enlarged Rab5a-positive endosomes (Figure 3I). Considering that Rab7a is recruited to Rab5a-positive endosomes by the PI3P-dependent mechanism of the Rab5-to-Rab7 switch [51], whereas Rab9a is delivered to EEs from TGN at a stage before the Rab5-to-Rab7 transition [52], we concluded that PI3P depletion delays the maturation of both subsets of LEs. Overall, this analysis suggests that Tf-loaded Rab5a-positive endosomes of PI3P-depleted cells are retained at the stage prior to the Rab5-to-Rab7 transition and have not lost the ability to take up membranes from TGN.

Taken together, these data indicate that PI3P depletion leads to the accumulation of recycling cargo (Tf) in Rab5a/Rab4 endosomes with maturation arrest toward the degradative pathway, which retains direct recycling capacity to the PM but cannot deliver recycling cargo to the indirect Rab11a-dependent recycling system.

### 3.5. PI3P Depletion Reorganizes the Pericentriolar Recycling System

Since the Rab11a-dependent pathway seems to be strongly affected, we further investigated the kinetics of ERC alteration after PI3P depletion. The ERC is defined as a heterogeneous mixture of endosomes compacted around the cell center [53,54] and includes at least two subsets: Rab11a and Arf6/Rab8a [55]. Therefore, we first analyzed the TfR-loaded and Rab11a-positive subdomain of ERC, which condenses around centrosomes and associates with several centrosomal proteins, including *γ*-tubulin [56]. In control cells, *γ*-tubulin staining displayed centrosome as a single punctate structure at the cell center surrounded by Tf-loaded endosomes (Figure 4A). However, in PI3P-depleted cells, internalized Tf did not concentrate around centrosomes but accumulated in enlarged perinuclear endosomes that were separated from the *γ*-tubulin-positive puncta and vacuolated over time (Figure 4A). A similar pattern was observed after Rab11a staining (Figure 4B). In control cells, the juxtanuclear Rab11a-positive vesiculo-tubular clusters of membranous organelles were arranged around the centrosomes (Figure 4B, 0 min). In ~20% of cells, it also showed subnuclear localization together with centrosomes (Figure 4B, 0 min, arrows), indicating a division of the ERC into juxtanuclear and subnuclear domains, reminiscent of that described for the intermediate compartment [57].

To analyze the distribution of Rab11a-positive organelles, we divided the cell into three zones (Figure 4C): the juxtanuclear area (JN), defined as the perinuclear area extending ½ nuclear diameter at a 90-degree angle from the nuclear center across the centrosomes; the subnuclear area (SN), defined as the outline of the nucleus using transmission images; and the peripheral area (p), all cytoplasm outside the JN area. Analysis of Rab11a staining at 10 min intervals (Figure 4B) showed that ERC change was rapid after PI3P depletion. At 10 min, ERC shifted to the SN area in many cells (46% versus 20% immediately after the addition of IN1; Figure 4D), ranging from a partial shift with centrosome split to a complete shift (Figure 4B, arrows). The displacement was observed in many cells after 5 min (Appendix A), and after 10 min resulted in a substantial relocation of Rab11a fluorescence (Figure 4E) to the SN area (from ~13% to ~49%) and a decrease in the JN area (from ~50 to ~21%). These relocations were quite dynamic as, after 20 min, Rab11a staining in the SN area decreased (Figure 4B), falling to ~20% (Figure 4E). However, this decrease was not associated with the return of Rab11a to the JN domain but with the appearance of cytoplasmic Rab11a structures (Figure 4B, inset), resulting in ~60% of Rab11a fluorescence being in the peripheral area of the cell (Figure 4E). Since the centrosomes remained in the SN area and no connection between the SN domain of the ERC and the Rab11a-positive cytoplasmic structures was detected, these data suggest a depletion of the SN domain and a scattering of Rab11-positive organelles in the cytoplasm. These organelles continued to grow and vacuolize 40 and 60 min after PI3P depletion (Figure 4B), resulting in an almost complete loss of Rab11a from the SN area (10–11%) and relocation of up to 75% to the peripheral regions of the cell (Figure 4E). Centrosomes shifted back to the JN area and did not associate with the majority of Rab11a-positive structures (Figure 4B). The distribution pattern of Rab11a-positive organelles persisted 120 min after PI3P depletion (Appendix A) and was maintained over the long term (i.e., after 24 h; data not shown). These data demonstrate that PI3P depletion has a profound effect on Rab11a endosome biogenesis and, consequently, on the stability of the Rab11a subdomain of the ERC. 

The observed dynamics of ERC redistribution and the appearance of peripheral Rab11a-positive vesicles could be reproduced under different conditions, as shown by the simultaneous staining of EEA1 and Rab11a in Appendix A. These effects were specific for the two Vps34-specific inhibitors IN1 and SAR405, but not for broader PI3K inhibitors LY294002 and wortmannin, and were observed in both Balb 3T3 and HeLa cell lines (Appendix A).

Because Rab11a analysis indicated relocation and depletion of the ERC-Rab11a subdomain, we monitored antibody-labeled TfRs and Rab11a simultaneously (Figure 5). Cell surface TfRs were labeled with mAbs at 4 °C and internalized for 60 min to establish the steady-state cycling [45]. Under these conditions, approximately 50% of Tf localized to the juxtanuclear area with Rab11a (Figure 5A,B, 0′). In contrast to Tf-AF^488^ labeling conditions (Figure 4A), which results in loss of Tf-AF due to recycling, continuous cycling of mAb-labeled TfRs allowed prolonged monitoring of the TfR-loaded subdomain of the ERC after PI3P depletion without a significant decrease in fluorescence signal. After 10 min of IN1 treatment, Rab11a remained with mAb/TfR and relocated to the SN area in half of the cells (Figure 5A, 10′). However, after 10 min, a significant amount of mAb/TfR was found in endosomes outside the JN area (Figure 5A, 10′), consistent with mAb/TfR cycling and accumulation in EEs, leading to reduced colocalization of mAb/TfR with Rab11a (Figure 5B, 10′). After 20 min of IN1 treatment and later, most Rab11a disappeared from JN and SN localization and appeared in peripheral vesicles (Figure 5A, 20′–120′) that did not largely overlap with mAb/TfR-loaded endosomes (Figure 5B, 20′–120′). Although large Rab11a- and mAb/TfR-positive entities were segregated, 3D colocalization analysis still identified 20–30% of pixel overlap (Figure 5B, 20′–120′). In zoomed images and 0.3 µm sections, small mAb/TfR- and Rab11a-positive entities were found near large Rab11a- and mAb/TfR-positive endosomes (Figure 5C). Three-dimensional reconstruction of 0.3 µm Z series (Figure 5C) revealed the segregation of Rab11a- and mAb/TfR-loaded domains, including segregation of mAb/TfR-loaded domains from large Rab11a-positive endosomes and Rab11a-domains from large mAb/TfR-positive endosomes. These data suggest that PI3P depletion induces segregation of Rab11a domains from recycling cargo-loaded endosomes and prevents the loading of Rab11a-positive carriers into the ERC.

Redistribution of Rab11a-positive endosomes was reversed after IN1 washout (Figure 4F), coinciding with the recovery of PI3P and reassociation of EEA1 with EEs (Figure 1E). As early as 2 min after washout, Rab11a endosomes began to concentrate within the Ø10 μm pericentriolar circle, further concentrating around centrosomes over the next 5 and 10 min and forming the typical juxtanuclear configuration after 30 min (Figure 4F,I). Pericentriolar concentration was associated with fragmentation of peripheral Rab11a-positive endosomes, suggesting microtubule-dependent transport toward the cell center. When ciliobrevin D (Cib D) was present during ERC recovery, a cytoplasmic dynein inhibitor [58] that completely dispersed the ERC within one hour (Figure 4G), Rab11a endosomes remained fragmented outside the Ø10 µm pericentriolar area. They did not concentrate around the centrosomes (Figure 4H,I), although a small cluster of pericentriolar Rab11 endosomes was identified after 10 min (Figure 4H). These results suggest that peripheral Rab11a endosomes of PI3P-depleted cells rapidly acquire a transport capacity toward the cell center after the restoration of PI3P production.

Since Rab11a can be activated at TGN-derived membranes [59,60], we also performed a colocalization analysis of Rab11a with the Golgi markers. Minimal overlap of Rab11a with GM130 and GS-15, cis and medial Golgi markers, and an expected moderate overlap of Rab11a with Vti1a and TGN38, two TGN markers, was observed in both control and IN1-treated cells (Appendix A). However, most of the peripheral Rab11a-positive endosomes, which are characteristic of PI3P-depleted cells, did not exhibit any of these TGN markers, suggesting that the peripheral Rab11a-positive endosomes are not the TGN-derived membranous organelles.

### 3.6. IN1 Treatment DEPLETES the Rab8a-Positive Subset of ERC

Several studies suggest that the pericentrosomal recycling system consists of at least Rab11a- and Arf6/Rab8a-positive subsets of endosomes [61]. Therefore, we next examined the effects of PI3P depletion on the integrity of the Rab8a-positive subset. Rab8a-positive membranes concentrated in the juxtanuclear area in control cells and were approached by internalized Tf (Figure 6A), resulting in approximately 50% colocalization (Figure 6B). In PI3P-depleted cells, staining for Rab8a was significantly reduced (Figure 6A,C), Rab8a-positive endosomes were barely detected in the pericentriolar area, and Rab8a-positive endosomes did not appear in the cell periphery (Figure 6A), as was observed for Rab11a-positive endosomes. These data indicate that PI3P is required for the biogenesis of the Rab8a-positive subset of ERC. Furthermore, these data suggest that the Rab8a subset is downstream of the Rab11a-positive subset during maturation.

### 3.7. Rab11a-Positive Endosomes Acquire Rab11-FIP5 However, Not Rab11-FIP3

To gain insight into the biogenesis of Rab11a-positive endosomes in PI3P-depleted cells, we analyzed whether they acquire Rab11-FIP5 (FIP5) and Rab11-FIP3 (FIP3) proteins. FIP5 recruits to Rab11 endosomes at earlier stages of EE maturation than FIP3 [62]. Both FIPs were present at perinuclear endosomes in control cells and colocalized with Rab11 in the pericentriolar area (Figure 7A). Outside the pericentriolar area, both FIPs were located in endosomal structures lacking Rab11a (Figure 7A), consistent with their Rab11a-independent membrane recruitment [63]. In PI3P-depleted cells, the enlarged and vacuolized Rab11a-positive endosomes were enriched in FIP5 and depleted of FIP3 (Figure 7A), resulting in a significant increase in colocalization of Rab11a with FIP5 and a decrease in colocalization with FIP3 (Figure 7B). Consistent with the previous data, FIP5-positive endosomes were not loaded with internalized Tf-AF^488^ and did not colocalize with Rab5a (Figure 7C and Appendix A). These data suggest that Rab11-positive endosomes are developed by PI3P-depleted cells at earlier stages of EE maturation and fail to acquire FIP3.

Considering that FIP3 is recruited to Rab11a endosomes at later stages of their maturation and contributes to their binding to dynein [64], the failure of FIP3 acquisition by Rab11a-positive endosomes in PI3P-depleted cells could prevent their migration to the pericentriolar area. To investigate this possibility, we monitored the reestablishment of the pericentriolar cluster of Rab11a/FIP5 endosomes 10 and 30 min after IN1 washout. As shown in Figure 7D, Rab11a-positive endosomes remained associated with FIP5 and concentrated in the pericentriolar area as early as 10 min, keeping the same extent of colocalization (Figure 7E). However, these endosomes acquired FIP3, resulting in a gradual increase in colocalization of Rab11 with FIP3 (Figure 7E). These data suggest that the migration of Rab11a/FIP5 endosomes toward the cell center is accompanied by the acquisition of FIP3 and that this process is dependent on the production of PI3P at EEs.

### 3.8. Segregation of Endosomal PI3P by Expression of PI3P-Binding Modules Alters Endosomal Trafficking of TfR

Given that pharmacological PI3P depletion has shown a change in EE maturation toward the ERC, we examined whether a similar effect can be induced by overexpression of PI3P-binding modules (EGFP-2xFYVE and YFP-p40PX) that sequester endosomal PI3P, thereby showing dominant-negative effects [34,65]. We either labeled cell surface TfRs with Tf-AF^555^ (Figure 8A) or anti-TfR mAb (Figure 8B) at 4 °C and analyzed them after 45 min of internalization at 37 °C or used the Tf-AF^555^ feeding protocol (Appendix A) to visualize the recycling circuit in 2xFYVE and p40PX expressing cells. Analysis of transfected cell samples allowed the simultaneous monitoring of the TfR pathway in transfected and untransfected cells (Figure 8, cells without green fluorescence). As a control, we used constructs expressing p40PX^R57Q^ and 2xFYVE^C215S^ modules with a mutation in the PI3P binding sites, as well as vectors expressing EGFP or YFP alone. 

In cells strongly expressing 2xFYVE and p40PX, both Tf-AF^555^ (Figure 8A) and mAb/TfRs (Figure 8B) accumulated in enlarged FYVE- and p40PX-positive endosomes and were highly colocalized there (Figure 8C). The typical JN and SN TfR accumulation that occurred in nontransfected cells (Figure 8A,B, arrowheads) was absent in 2xFYVE- and p40PX-expressing cells, even if the expression level was lower (Appendix A), suggesting that the expression of PI3P-binding modules prevents TfR loading into the ERC in the same manner as short-term PI3P depletion by the Vps34 inhibitor. Both Tf-AF^555^- and mAb/TfRs were frequently observed outside 2xFYVE- and p40PX-positive enlarged endosomes in tubular and small punctate forms (Figure 8A,B), suggesting that TfRs can be segregated into tubular EEs that form a tubular endosomal network of the EE recycling circuit. In cells transfected with vectors expressing PI3P-binding-deficient modules, 2xFYVE^C215S^ (Figure 8D) and p40PX^R57Q^ (Appendix A), and with vectors expressing EGFP (Figure 8D) or YFP (Appendix A) fluorescent tags, internalized TfRs loaded JN and SN endosomes as in nontransfected cells. These data suggest that the effect of 2xFYVE- and p40PX expression is specific to their PI3P-binding capacity and was not caused by transfection conditions.

As shown in the short-term PI3P-depleted cells (Figure 2D), TfRs recycled with indistinguishable kinetics from 2xFYVE- and p40PX-expressing cells and nontransfected cells (Figure 8E), indicating that 2xFYVE- and p40PX-decorated endosomes maintain the capacity of recycling to the cell surface and all recycling-associated downstream functions, such as swollen endosomes from IN1-treated cells. Similar results were also obtained from the imaging analysis of TfR recycling (Appendix A).

The enlarged endosomes of 2xFYVE- and p40PX-expressing cells were Rab5a-positive, and the expression of 2xFYVE^C215S^ or EGFP did not alter the Rab5a distribution pattern (Figure 9A), suggesting PI3P-specific enlargement of EEs by saturation of PI3P. A significant proportion of Rab5a did not colocalize with 2xFYVE (Figure 9C) and p40PX (Figure 9D), indicating that PI3P-binding module expression affects a subdomain of Rab5 endosomes. Given that internalized TfRs were localized in 2xFYVE- and p40PX-positive structures (Figure 8A,B), we concluded that these PI3P-enriched domains are responsible for cargo recycling. These endosomes were also positive for Rab9a (Appendix A), with similar colocalization to Rab5a (Figure 9C,D), as observed after short-term PI3P depletion (Figure 3I,J), suggesting a similar alteration of EE maturation toward LE. In contrast to short-term PI3P-depleted cells, no significant accumulation and colocalization of APPL1 were observed in cells expressing PI3P-binding modules (Appendix A).

As observed with the short-term PI3P-depleted cells, very little Rab11a was found in association with enlarged endosomes of 2xFYVE- and p40PX-expressing cells (Figure 9B–D). Rab11a-positive structures were scattered in punctate structures at the cell periphery (Figure 9B, arrows) and did not form JN or SN cluster as did nontransfected, 2xFYVE^C215S^-, the and EGFP-transfected cells (Figure 9B, arrowheads). These data suggest that cells expressing PI3P-binding modules do not develop the Rab11a-positive subdomain of the ERC. In addition, none of the known markers of other subdomains within ERC, such as Rab35/ARF6, Rab8, Rab10, and Evectin-2, were observed in the pericentriolar area of PI3P-binding module-expressing cells (data not shown).

Overall, Tf trafficking analysis and endosome phenotyping in PX- and 2xFYVE-transfected cells suggest that the overexpression of PI3P-binding modules has a similar effect on the endosomal recycling circuit as PI3P depletion. It does not affect endocytic uptake of TfRs into EEs, EE fusion, sorting of recycling cargo, and recycling from EEs to PM. Nevertheless, it does affect endosomal maturation in the EE-to-ERC transition phase. 

## 4. Discussion

This study investigated the contribution of PI3P to the physiology of the EE system. Here, we show that rapid depletion of Vps34-derived PI3P production by the Vps34-specific inhibitor discontinues PI3P-dependent functions of EEs and rearranges EEs into a new, PI3P-independent functional configuration. This rearrangement is promptly established and is maintained over the long term. Analysis of the new configuration shows that Vps34-derived PI3P is not essential for the fast recycling from EEs but significantly contributes to the maturation of the vacuolar EE domain toward the degradation circuit. The alterations of the EE vacuolar domain functions were demonstrated after short-term pharmacological [16,21,25,36,65,66,67] and genetic [21] PI3P depletion, as well as after long-term knockdown [16,17] and knockout [18,19] of Vps34. Our study extends the characterization of the PI3P contribution in the vacuolar EE domain functions. However, our study also demonstrates that Vps34-derived PI3P is essential for the establishment of the Rab11-dependent pathway, including the sorting of recycling cargo in this pathway and membrane flux from EEs to ERC. Thus, Vps34-derived PI3P is indispensable for the recycling circuit to maintain the slow recycling route and thus ERC biogenesis.

Physiological processes at EEs use PI3P as a platform for the recruitment of effector proteins that drive EE maturation [3,4,6]. Nearly 80 cellular proteins use either FYVE or PX domains to bind PI3P [4,68]. Some of these are strictly dependent on PI3P availability and rapidly dissociate from EEs following PI3P depletion, leading to rapid discontinuation of PI3P-dependent processes. The release of FYVE- and PX-domain-containing proteins from EEs also opens the space for the recruitment of PI3P-independent effector proteins. An example of this is the Hrs retention and Rab5-dependent recruitment of APPL1 to the EEs of PI3P-depleted cells (Figure 1 and Figure 3), two proteins that are involved in the earliest stages of the EE tract [21,69]. APPL1 backconversion is a striking feature of acutely PI3P-depleted cells, as was also observed after the acute activation of MTM1 phosphatase at EEs [21]. This was not observed after the long-term sequestration of PI3P by FYVE- and PX-binding modules (Figure 9). Accordingly, the quick establishment of new functional configuration after pharmacological PI3P-depletion and rapid restoration of the PI3P-dependent configuration after reversal of PI3P production allows the study of EE plasticity and EE physiological pathways without the long-term perturbation of the cell.

### 4.1. PI3P-Dependent Maturation of Vacuolar EE Domain

The salient feature of enlarged Rab5 endosomes in PI3P-depleted cells is a delay in maturation in both EE-to-LE routes, characterized by the conversion of Rab5 domains into either Rab7 or Rab9 domains [2]. The maturation of the Rab5 domain requires the Vps34-derived PI(3)P to switch off Rab5 activity and knockdown of Vps34 [17,18,70]. Prolonged activity of Rab5 following the expression of constitutively active mutant [71] results in a similar delay in maturation and enlargement of EEs. The knockout study suggests direct negative feedback between Vps34-derived PI(3)P and Rab5 activity through PI(3)P-dependent recruitment of Rab5-GAP [17,70]. Thus, the reduced recruitment of Rab7a to enlarged Rab5 endosomes in IN1-treated cells (Figure 3) indicates a delay in the conversion of Rab5 to Rab7. This step also requires PI3P-dependent recruitment of the SAND1/MON1–CCZ1 complex that promotes the Rab switch [51], the release of the multisubunit class C core vacuole/endosome tethering complexes (CORVET), and the recruitment of the homotypic fusion and protein sorting complexes (HOPS) [72,73]. In contrast to Rab5-Rab7 conversion, which relies on the recruitment mechanism, conversion to the Rab9 domain relies on the influx of Rab9 membranes into EEs from the TGN. The influx occurs at later stages of EE maturation, prior to the initiation of Rab5-Rab7 conversion [49,52], followed by inactivation of Rab5 by a mechanism possibly involving Rab5-GAP SGSM3 [74] and the development of the Rab9 LE domain, which is distinct from the Rab7a domain [50]. The persistence of Rab9a in a subdomain of the enlarged Rab5 endosomes in PI3P-depleted cells (Figure 3) suggests that the Rab5-to-Rab9 conversion or the downstream segregation of Rab9 membranes requires PI3P-dependent functions.

### 4.2. PI3P-Dependent Cargo Sorting

This study analyzed the trafficking and sorting of Tf/TfR, a clathrin-dependent cargo (CDE) known to be recycled by EEs via multiple pathways, including the Rab11 pathway that loads the ERC for slow recycling [75]. It is generally assumed that TfRs, upon entry into the EEs, undergo the geometry-based sorting process, also known as bulk recycling [44]. Our study shows that in PI3P-depleted or PI3P-saturated cells, Tf is removed from the cell with indistinguishable kinetics as in control cells, indicating that the geometry-based sorting process can function in the absence of PI3P. However, Tf-TfR sorting into Rab11 carriers that transport them to the pericentriolar ERC requires PI3P-associated functions.

Because the geometry-based sorting process operates in PI3P-depleted cells, the lack of TfR sorting could be related to processes that occur later in the maturation of Rab11 vesicles. PI3P depletion dissociates most of the PX domain-containing endogenous SNXs from EEs, including SNX1 and SNX3 (Figure 1H), which may be required for TfR loading into Rab11 endosomes. Although these SNXs are mainly involved in sequence-dependent recycling processes [39], they might also be involved in TfR recycling. SNX1 induces tubule formation in EEs [40] and is required for rapid FERARI-dependent Tf/TfR loading into Rab11 carriers at EEs [41]. Therefore, the absence of SNX1 at this stage of Rab11 vesicle maturation could be responsible for the lack of TfR loading into Rab11 vesicles. This would imply that Rab11 endosomes are formed as empty vesicles that require PI3P-dependent docking at EEs to take up cargo and perhaps lose the ability to fuse and gain the ability to minus-end directed migration. 

In addition to SNX1, the absence of SNX3 may also be related to the defective TfR sorting into Rab11 endosomes. Endogenous SNX3 is abundant at EEs of control Balb 3T3 cells, binds specifically to PI3P [76], and may be associated with retromer-mediated cargo sorting to TGN [77]. However, SNX3 can also physically interact with TfR along with Vps35 and form a non-classical retromer complex that sorts dimerized TfRs into recycling endosomes [42] for retromer-independent recycling [43]. Therefore, sequence-dependent sorting may be required for Tf-TfR loading into the Rab11-dependent recycling route. In the absence of this process, as shown here in PI3P-depleted cells (Figure 2) and in SNX3 knockdown cells [42], TfRs can be sorted in the bulk pathway and recycled back to the cell surface.

### 4.3. PI3P-Independent Biogenesis of Rab11 Endosomes

Our data show that Rab11 endosome biogenesis does not depend on the presence of PI(3)P at EEs. The available data from the literature do not explain how and where Rab11 is bound to membranes and activated. Many studies demonstrated Rab11 at EEs (rev. by [59]); therefore, it is reasonable to expect that Rab11 is recruited to and activated at EE membranes. In addition, Rab11 can be recruited to and activated at membranes of the late Golgi and TGN [60]. However, the available data do not establish a molecular link for Rab11 activation at EEs. Indeed, EEs are characterized by the presence of Rab5, which can recruit known Rab11 GAP proteins and thereby prevent its activation [74], but it is unknown that Rab5 recruits Rab11 GEFs. Additionally, Rab7a and Rab9a that join later stages of EE maturation can recruit GAPs to switch off but not GEFs [74] to switch on Rab11 activation. Therefore, Rab11 activation at EEs may be expected in an environment devoid of Rab5, especially since it is also unknown whether Rab11 can recruit Rab5 GAP. Several studies suggest that Rab11 can be activated downstream of Rab4 [40,78].

In addition to Rab11 activation, insufficient information about the molecular machinery and EE differentiation stage that facilitates Rab11 recruitment to EEs is available. The absence of Tf in Rab11 endosomes of PI3P-depleted cells indicates that Rab11 endosomes may be generated as recycling-cargo empty vesicles and loaded with the recycling cargo later, consistent with the kiss-and-run model described by Solinger et al. (2020) [41]. The kiss-and-run events, which occur rapidly and frequently at SNX1-positive tubular domains of EEs, may explain the overlap of Rab11 with EE markers observed in many studies. Although Rab11 FIPs are considered downstream effectors of Rab11 [79], FIPs may facilitate Rab11 recruitment at distinct stages of EE maturation since FIPs can be recruited to EE membranes independently of Rab11 [63]. Our study shows that peripheral Rab11-positive compartments arising in PI3P-depleted cells are highly enriched in FIP5 and contain little FIP3 (Figure 7). This also indicates that Rab11 vesicle budding and FIP5 recruitment are PI3P-independent processes. Based on the Tf loading kinetics, Tf was proposed to enter the Rab11-dependent recycling endosomes via FIP1A- and FIP2/FIP5-dependent pathways and proceeds to pericentriolar compartments through sequential loading of FIP1B, FIP1C, and FIP3 compartments [62]. Therefore, Rab11 vesicles are likely to be formed by PI3P-depleted cells at the proximal stages of EE maturation before they acquire transport competence to the ERC.

### 4.4. PI3P-Dependent Maturation of the Rab11-Dependent Pathway

A hallmark of PI3P depletion is the peripheral accumulation of enlarged Rab11 endosomes that vacuolate over time. The enlargement may be related to the disruption of several PI3P-dependent processes. The prolonged retention of homotypic fusion capacity may explain the enlargement of the Rab5-positive vacuolar domain and may be related to the enlargement of EE-derived Rab11 endosomes. Although EEs of PI3P-depleted cells lose EEA1, an important tethering factor required for their homotypic fusion [36], they can maintain their fusion capacity through CORVET tethering complexes [72,73]. Their replacement with HOPS complexes is essential for the maturation of the vacuolar domain into LEs and requires PI3P-dependent functions [73]. On the other hand, Rab11 vesicle formation also requires the conversion or replacement of CORVET to avoid the fusion of Rab11 vesicles with EEs. The fusion ability of Rab11 carriers could be maintained if the replacement of CORVET in the recycling pathway occurs before PI3P loss from endosomes and requires PI3P-associated functions. However, this relationship was not demonstrated in the literature, and the tethering complexes dynamic in the EE recycling circuit was not fully elucidated. Recently, it was also shown that the CORVET-specific subunits Vps3 and Vps8 regulate the vesicular transport of Rab4-positive recycling vesicles from EE to RE, which is required for their fusion with Rab11 and CHEVI endosomes [80]. 

In addition to the prolonged tethering capacity, the enlargement of the EE vacuolar domain can be explained by the depletion of PI(3,5)P2 production and acidification of EEs. The PI(3,5)P2 deficit in PI3P-depleted cells could be due to the lack of substrate (PI3P) and decreased recruitment of PIKfyve, an enzyme that produces it [81]. Inhibition of these processes would prevent the recruitment of the downstream machinery required for reverse budding [82], leading to accumulation of limiting membranes and enlargement of EEs. Furthermore, both PI(3,5)P2 depletion and Vps34 suppression can impair the assembly of vacuolar proton-translocating ATPases (V-ATPase) [81]. These alterations are most likely to be found in the vacuolar EE domain but cannot explain the enlargement and swelling of Rab11 endosomes after their formation in the tubular EE domain.

The enlargement of Rab11 endosomes in PI3P-depleted cells could be a combined effect of a sustained homotypic fusion capacity and a defective minus-end directed movement capacity. Depletion or suppression of many factors that drive minus-end motility of Rab11 endosomes leads to their dispersion and loss of pericentriolar accumulation, including flotillin [83], FIP3 [84,85], ASAP1 [85], DLIC-1 [64], and SNX4-dependent association with microtubules [86]. A similar effect was obtained by pharmacological inhibition of dynein with ciliobrevin D (Figure 4); [27]. The acquisition of minus-end migration capacity might be associated with the sequential recruitment of class I FIPs to endosomes. Recruitment of class II FIPs, including FIP5, links Rab11 endosomes to kinesins, ensuring plus-end motility, whereas recruitment of FIP3 links to dynein [62,64] and redirects their movement toward the minus-end. Thus, the absence of FIP3 at Rab11-positive endosomes of PI3P-depleted cells could facilitate their movement away from the cell center, as a prevalence of plus-end directed motors at these endosomes, thereby facilitating their homotypic fusion. After IN1 washout, PI3P activity at the endosomes is rapidly restored, and Rab11 endosomes begin migrating toward the cell center (Figure 4 and Figure 7), suggesting that acquisition of minus-end migratory capacity is a PI3P-dependent process. Acquisition of FIP3 may require a PI3P-dependent interaction of Rab11a/FIP5-positive endosomes during cargo uptake through the kiss-and-run flickering, as the knockout of FERARI and SNX1 resulted in enlarged Rab11 structures [41]. Moreover, the inhibition of minus-end directed motility could be an off-target effect of IN1 by acting on the motor machinery. However, the latter is not likely, as the same phenotype was observed after treatment with another Vps34 inhibitor (SAR405).

### 4.5. PI3P-Dependent Biogenesis of the ERC

Our study suggests the essential role of PI3P-associated functions in ERC biogenesis. In PI3P-depleted cells, Tf-loaded, Rab11/FIP5- and Rab8a-positive endosomes fail to accumulate in the pericentriolar area (Figure 4, Figure 5, Figure 6, Figure 7, Figure 8 and Figure 9). Despite the great importance of the ERC physiological function, surprisingly little is known about its biogenesis and maintenance [8,54]. The ERC is a heterogeneous mixture of subcompartmentalized vesicular and tubular endosomes that collect and sort CDE and clathrin-independent (CIE) cargo for recycling to the PM or exchange cargo with the TGN [61]. In addition to Rab11, which defines the ERC [87,88,89], membranous entities in the pericentriolar area are enriched in Rab8, Rab10, Rab12, Rab14, Rab17, Rab22a, Arf6, and Rab35 [2]. These GTPases are activated at ERC endosomes in complex cascades [90], indicating high diversification of membrane maturation pathways in the ERC. In general, the ERC membranous entities can be divided into two subsets, Rab11 and Arf6/Rab8 [55], which are reported to be associated with CDE and CIE recycling pathways, respectively [75]. Although the Rab11a and Rab8a pathways may be different and involve separate membrane entities, they share many elements of the activation cascade and effector functions [61,91]. Thus, it is likely that Rab8 acts downstream of Rab11 in the maturation of the Rab11 subset and contributes to TfR recycling. This interpretation implies the existence of a Rab11-to-Rab8 cascade at the Rab11 subset, which is supported by the evidence that Rab8 can be activated by Rab11 together with Rab10 [91,92]. Therefore, the absence of Rab11 vesicle trafficking in PI3P-depleted cells would lead to the disappearance of the Rab8 subset, as shown in our study (Figure 6), suggesting that Vps34-derived PI3P is required for the biogenesis of both subsets of ERC.

The lack of incoming flow of Rab11a-positive and TfR-loaded endosomes into the pericentriolar area enabled us to also monitor the outgoing flow emanating from the ERC (Figure 4). The kinetics of the Rab11a/TfR domain translocation in PI3P-depleted cells suggests that the ERC may also be organized into spatial subdomains as described for the intermediate compartment (IC) [57]. The pericentriolar IC subdomain separates from the Golgi and translocates from the JN to the SN area before delivering biosynthetic cargo to the PM, which is a similar sequence observed for the ERC Rab11a/TfR domain in PI3P-depleted cells. Thus, with the discontinued supply of Rab11 carriers, outgoing flow from the Rab11a/TfR domain to the PM empties the ERC. We can speculate that there is a feedback mechanism between the peripheral Rab11 endosomes and the ERC. In the absence of the ERC after microsurgical removal [93], Rab11 endosomes coalesce and fuse in the peripheral cytoplasm, as observed in PI3P-depleted cells.

### 4.6. Possible off-Target Effects of VPS34-IN1

In this study, we used a 3 µM concentration of IN1, which is known to be a highly specific Vps34 inhibitor. At this concentration, IN1 can inhibit the activity of PIP5K1a and PI5K1c lipid kinases by 30–40% in an in vitro assay [22]. These kinases convert PI4P into PI4,5P2. A PI5K1c (PIPKIγ) splice variant PIPKIγi5 was found on EEs and may contribute to cargo sorting toward LEs and regulate EE maturation by recruiting SNX5 and Rab7a [94]. Both PIP5K1a and c are found at endosomal tubules and can recruit PI4,5P2-dependent SNXs (SNX5, SNX6, and SNX9) to facilitate tubule formation [94]. Moreover, 1 µM IN1 inhibited several protein kinases among 300 kinases tested [22]. Thus, although it is not very likely, we cannot rule out the possibility that IN1 in Balb 3T3 cells influences the observed effect on Rab11 recruitment and endosome vacuolization.

## 5. Conclusions

Endosomal recycling processes are crucial for many pathophysiological states, such as the biological behavior of tumor cells [95] or the assembly and egress of viruses [96,97]. Despite the growing knowledge of the physiology of endosomal recycling and ERC, many aspects of these processes remain to be elucidated. Our study focuses on the role of endosomal PI3P and shows that TfR can be recycled from EEs in the absence of PI3P, but it cannot be sorted into the Rab11-dependent slow recycling route in the absence of PI3P-dependent functions. Our study also shows that Vps34-derived PI3P is not required for the development of Rab11/FIP5-positive recycling endosomes, but their maturation into FIP3-positive endosomes, loading with recycling cargo (TfR), and migration to the pericentriolar region to form the ERC require PI3P-associated functions. In light of recent studies [27,41,42,43] that have provided a new avenue for understanding these processes, we propose that PI3P-dependent recruitment of SNX1 and SNX3 plays an essential role in the loading of Rab11 endosomes with recycling cargo and their maturation into the ERC. Thus, our study suggests that PI3P is indispensable for the maintenance of the Rab11 pathway and ERC biogenesis. The use of rapidly acting and reversible inhibitor of PI3P production allows the study of these processes under endogenous expression. The next step would be to analyze the contribution of Rab11 FIPs and SNXs under more specific suppression conditions.

## Figures and Tables

**Figure 1 cells-11-00962-f001:**
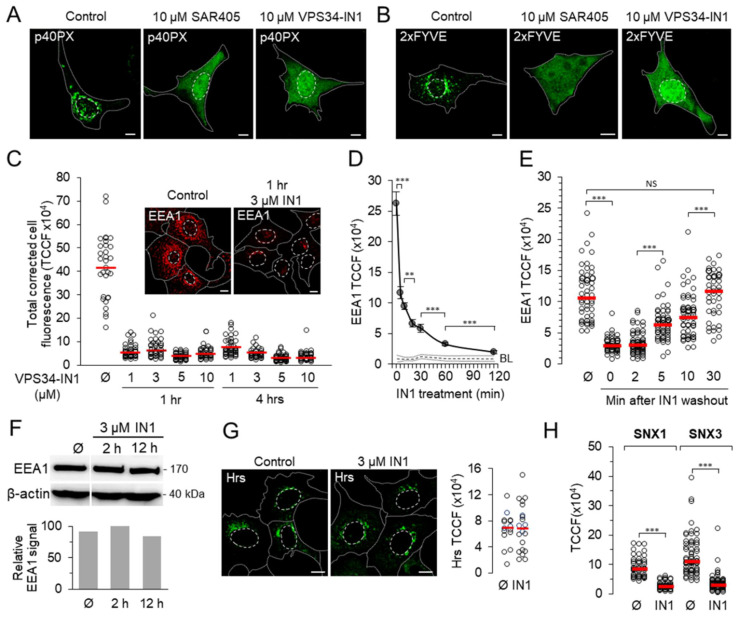
Pharmacological inhibition of Vps34 leads to the rapid dissociation of PI3P-binding proteins from EE membranes. (**A**,**B**) Dissociation of fluorescent PI3P-binding domains. Images of Balb 3T3 cells transfected with YFP-PX_P40phox_ MSCV (p40PX) or EGFP-2xFYVE_Hrs_ MSCV (2xFYVE) and 48 h after transfecton treated with either 10 μM VPS34-IN1 (IN1) or SAR405 for 4 h. Representative images of all samples (*n* = 12) are provided in the Appendix A. (**C**) Dissociation of EEA1 in cells treated with different concentrations of IN1 for 1 and 4 h. Fixed and permeabilized cells were stained against EEA1 and analyzed by confocal microscopy. Total corrected cell fluorescence (TCCF) of each cell was quantified using Fiji on the Otsu-thresholded confocal images. The data represent individual cells, and the horizontal bars represent the median value. The insets show a representative image of control and IN1-treated cells. Representative images of all samples are provided in Appendix A. Cell borders are indicated by fine dotted lines and cell nuclei by fine dashed lines. Bars, 10 μm. (**D**) Time course of EEA1 dissociation after addition of 3 µM IN1 to cells. The TCCF of EEA1 was determined as described in C. Data represent individual cells, and horizontal bars represent the median value; *** *p* ˂ 0.001, ** *p* ˂ 0.01 (one-way ANOVA). BL, background level. (**E**) Recovery of EEA1 staining on endosomes after IN1 washout. Cells were treated with IN1 for 120 min. The TCCF of EEA1 was determined as described in C. Data represent individual cells, and horizontal bars represent the median value; *** *p* ˂ 0.001(one-way ANOVA). Representative images of all samples are provided in the Appendix A. (**F**) Western blot analysis of EEA1 in control (Ø) and IN1-treated cells. Cropped blot images from the same membrane and densitometric quantification are shown. The original membrane is presented in Appendix A. (**G**) Images and TCCF intensity of Hrs in control (Ø) and 120 min IN1-treated (3 μM) cells. (**H**) Fluorescence signal intensity (TCCF) of SNX1 and SNX3 in untreated (Ø) and IN1-treated (120 min with 3 µM IN1) cells. Data represent individual cells, and horizontal bars represent median values. *** *p* ˂ 0.001 (two-tailed *t*-test). Representative images of all samples are provided in the Appendix A.

**Figure 2 cells-11-00962-f002:**
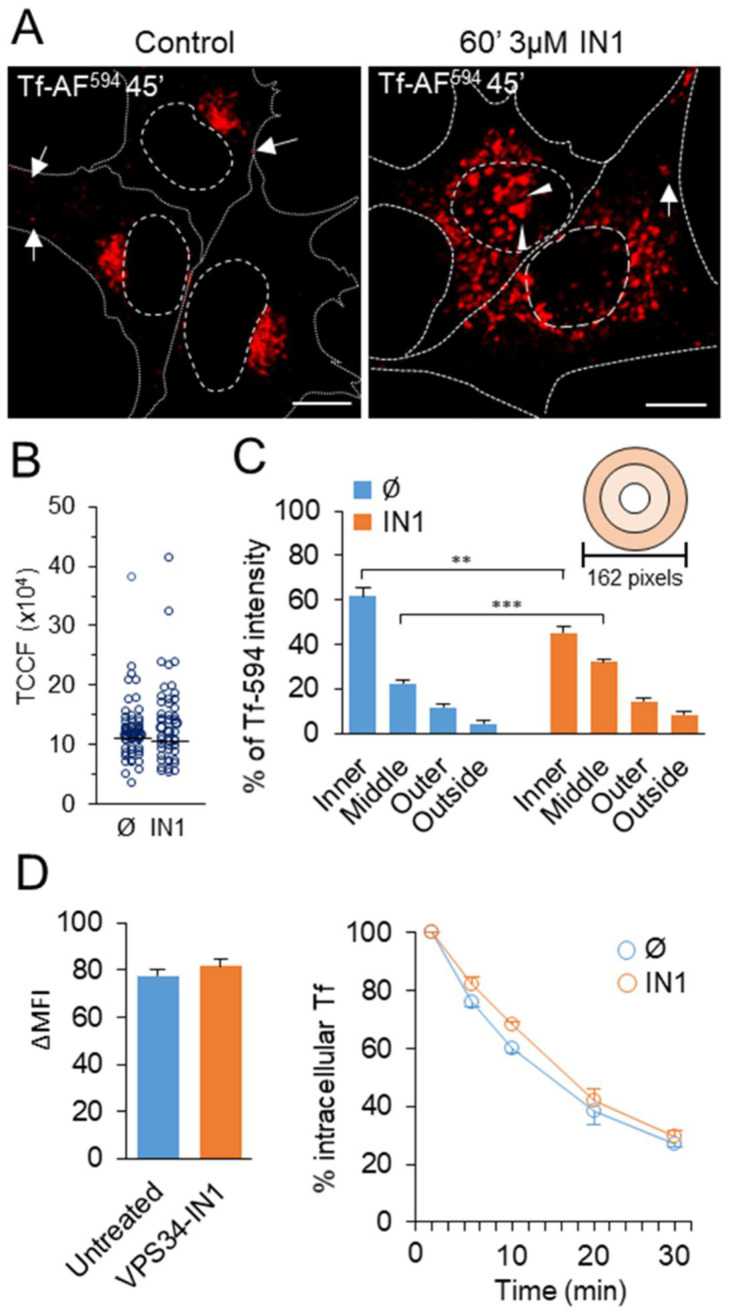
PI3P depletion reorganizes the early endosomal recycling circuit. Balb 3T3 cells were treated with either 3 μM VPS34-IN1 (IN1) or tissue culture medium containing DMSO (control) for 60 min and incubated with either Tf-AF^488^ or Tf-AF^594^ for 45 min in the presence of the inhibitor. (**A**) Representative confocal images (*n* = 8) of internalized Tf-AF^594^. Cell borders are indicated by fine dotted lines and cell nuclei by fine dashed lines. Bars, 10 μm. (**B**) Quantification of intracellular Tf-AF^594^ by Fiji analysis of images. The dots represent the TCCF of individual cells in the representative experiment (*n* = 4). (**C**) Quantification of Tf-AF^594^ distribution using Fiji. Concentric circles (0–54, 55–108, and 109–162 pixel distance) were centered on the area with the highest fluorescence signal. The intensity of Tf-AF^594^ within the circles was quantified relative to the total intensity in the whole cells. Data represent mean ± SEM; *** *p* ˂ 0.001, ** *p* ˂ 0.01 (one-way ANOVA). (**D**) Pulse-chase analysis of TfR recycling. Cells were pulsed with Tf-AF^488^, chased in a medium containing unlabeled Tf, and intracellular fluorescence quantified by flow cytometry. Fluorescence intensities (ΔMFI) after a 45 min pulse are shown on the left, and the kinetics of loss of intracellular fluorescence by TfR recycling is shown on the right. Data represent mean ± SEM from three independent experiments.

**Figure 3 cells-11-00962-f003:**
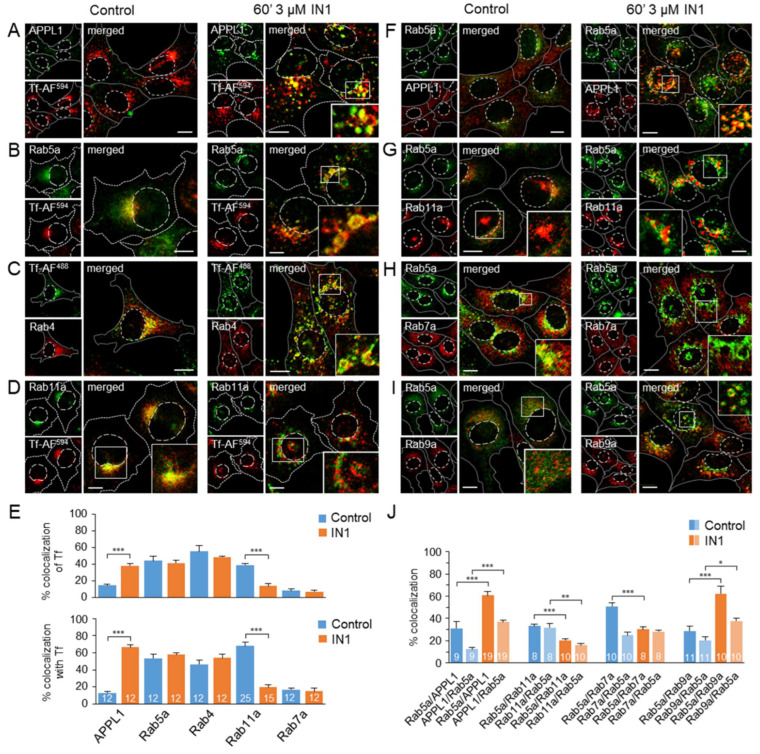
Co-localization analysis of internalized Tf (**A**–**E**) and Rab5a (**F**–**J**) in PI3P-depleted cells. (**A**–**D**) Colocalization of internalized Tf with APPL1, Rab5a, Rab4, and Rab11a in control and IN1-treated cells. Balb 3T3 cells were treated with either 3 μM VPS34-IN1 (IN1) or tissue culture medium containing DMSO (control) for 60 min and incubated with either Tf-AF^488^ or Tf-AF^594^ for 45 min in the presence of the inhibitor. Steady-state organelles were visualized on fixed and permeabilized cells using Ab reagents against cellular proteins that characterize the maturation steps of the membranous organelles. The antibody reagents used are listed in Appendix A, and each marker is described in Appendix A. Shown are the representative confocal images. The insets show a zoomed area box. Cell borders are indicated by fine dotted lines and cell nuclei by fine dashed lines. Bars, 10 μm. (**E**) 3D colocalization of Tf with organelle markers (upper panel) and organelle markers with Tf (lower panel) is based on the M1/M2 coefficients of pixel overlap measured over the Costes-algorithm thresholded z-stacks of confocal images. Data are mean ± SEM per cell (number of cells indicated in bars). Statistical significance was determined using Student’s t-test (*** *p* ˂ 0.001). (**F**–**I**) Representative images (*n* = 8–12) of Rab5a and membranous organelle markers in control and IN1-treated Balb 3T3 cells. Insets show zoomed area box. Cell borders are indicated by fine dotted lines and cell nuclei by fine dashed lines. Bars, 10 μm. (**J**) The 3D colocalization of Rab5a with organelle markers and organelle markers with Rab5a based on the M1/M2 coefficients of pixel overlap measured across the Costes-algorithm thresholded z-stacks of confocal images. Data are mean ± SEM per cell (number of cells indicated within bars). Statistical significance was determined using Student’s *t*-test (*** *p* ˂ 0.001; ** *p* ˂ 0.01; * *p* ˂ 0.05).

**Figure 4 cells-11-00962-f004:**
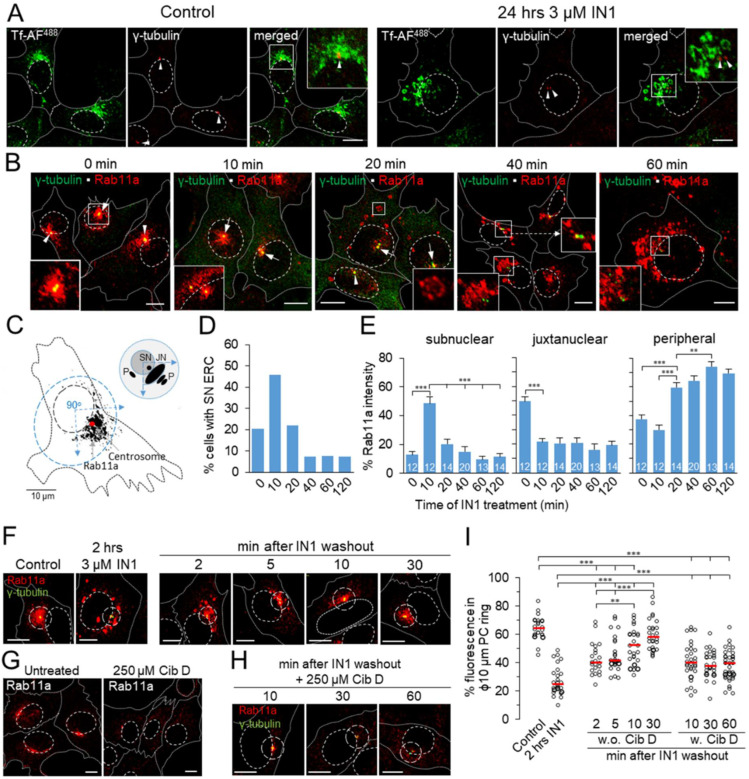
Reorganization of the pericentriolar recycling system in PI3P-depleted cells. (**A**) Representative images (*n* = 8–12) of 45 min internalized Tf-AF^488^ and *γ*-tubulin in control and 24 h IN1-treated Balb 3T3 cells. (**B**) Representative overlaid images of *γ*-tubulin and Rab11a in cells treated with IN1 for 0–60 min. Insets show zoomed area box. Arrowheads indicate juxtanuclear accumulation, and arrows indicate subnuclear Rab11a staining. (**C**) Subcellular Rab11a distribution zones relative to centrosomes on control cell images and schematic representation of subnuclear (SN), juxtanuclear (JN), and peripheral (*p*) areas used for ImageJ quantification of Rab11a distribution signal. (**D**) Percentage of cells with subnuclear ERC distribution after 0–120 min treatment with 3 µM IN1. (**E**) Quantification of Rab11a distribution across three zones in cells after 0–120 min treatment with 3 µM IN1. The intensity of Rab11a in the SN, JN, and *p* zones was quantified on the images using ImageJ relative to the total intensity in the whole cells. Data are mean ± SEM (number of cells indicated in bars); *** *p* ˂ 0.001, ** *p* ˂ 0.01 (one-way ANOVA). (**F**) Representative overlaid images of Rab11a distribution (red fluorescence) relative to centrosomes stained with γ-tubulin (green fluorescence) after 120 min of IN1-treatment, followed by IN1 washout and incubation in IN1-free medium for 2–30 min. Representative images of all samples are shown in Appendix A. The intensity of the Rab11a signal (TCCF) throughout the cell area and within the ϕ10 µm ring around the centrosomes (dashed circle) was analyzed using ImageJ on sections in the focal plane. (**G**) Representative images of endogenous Rab11a staining in control cells and cells treated with 250 µM ciliobrevin D (Cib D) for 60 min. (**H**) Representative overlaid images of Rab11a distribution relative to centrosomes (γ-tubulin) after 120 min of IN1-treatment, followed by IN1 washout and incubation in IN1-free medium with 250 µM ciliobrevin D for 0–30 min. Representative images of all samples are provided in the Appendix A. The intensity of the Rab11a signal throughout the cell area and within the ϕ10 µm ring around the centrosomes (dashed circle) was analyzed using ImageJ in sections on the focal plane. (**I**) The percentage of Rab11a signal within the pericentrosomal (PC) ϕ10-µm ring in 23–26 cells as described in (**F**) and 29–35 cells as described in (**H**). Data represent individual cells and horizontal bars represent median; *** *p* ˂0.001, ** *p* ˂ 0.01 (one-way ANOVA). Cell borders are indicated by fine dotted lines and cell nuclei by fine dashed lines. Bars, 10 μm.

**Figure 5 cells-11-00962-f005:**
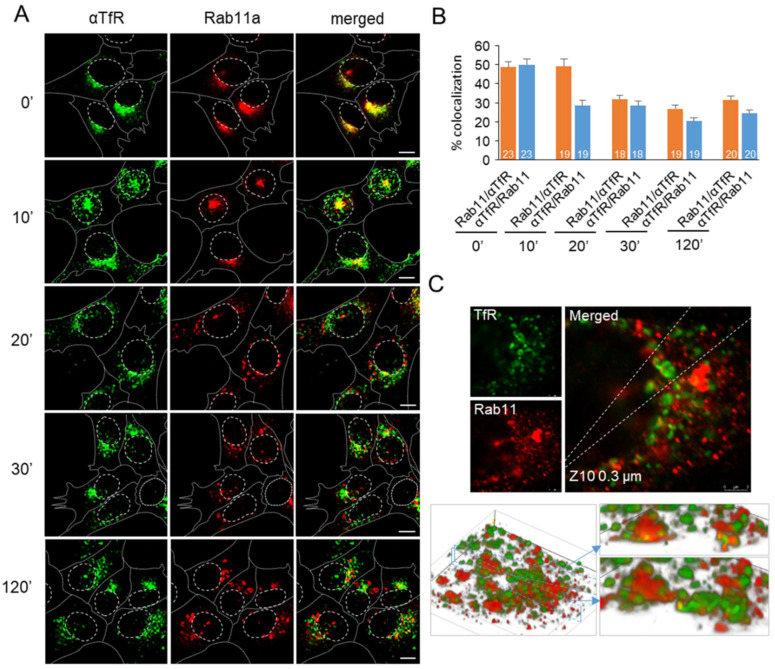
IN1 treatment splits recycling cargo from Rab11a-positive membrane domains. Cell surface TfRs of Balb 3T3 cells were labeled with anti-TfR mAb (R17) at 4 °C, labeled TfRs were internalized at 37 °C for 60 min to achieve the steady-state cycling, and cells were treated with 3 µM IN1. (**A**) Representative images (*n* = 15–20) of mAb-labeled TfRs and Rab11a before (0′) and 10′–120′ after treatment with IN1. Cell borders are indicated by fine dotted lines and cell nuclei by fine dashed lines. Bars, 10 μm. (**B**) The 3D colocalization of Rab11a and mAb-labeled TfRs based on the M1/M2 coefficients of pixel overlap measured across the Costes-algorithm thresholded z-stacks of confocal images. Data represent mean ± SEM per cell (number of cells indicated in bars). (**C**) Example of 3D reconstruction of Rab11a- and mAb/TfR-labeled endosomes after 120 min treatment with 3 µM IN1. The top panel shows confocal images (z10 of 0.3 µm sections), and the bottom panel shows 3D reconstruction of the entire z-stack (21 slices) using the Volume Viewer plugin. The image shown on the left presents the 3D view of the entire stack, and the images on the right show the view across the vertical slice through the stack as indicated by dashed lines.

**Figure 6 cells-11-00962-f006:**
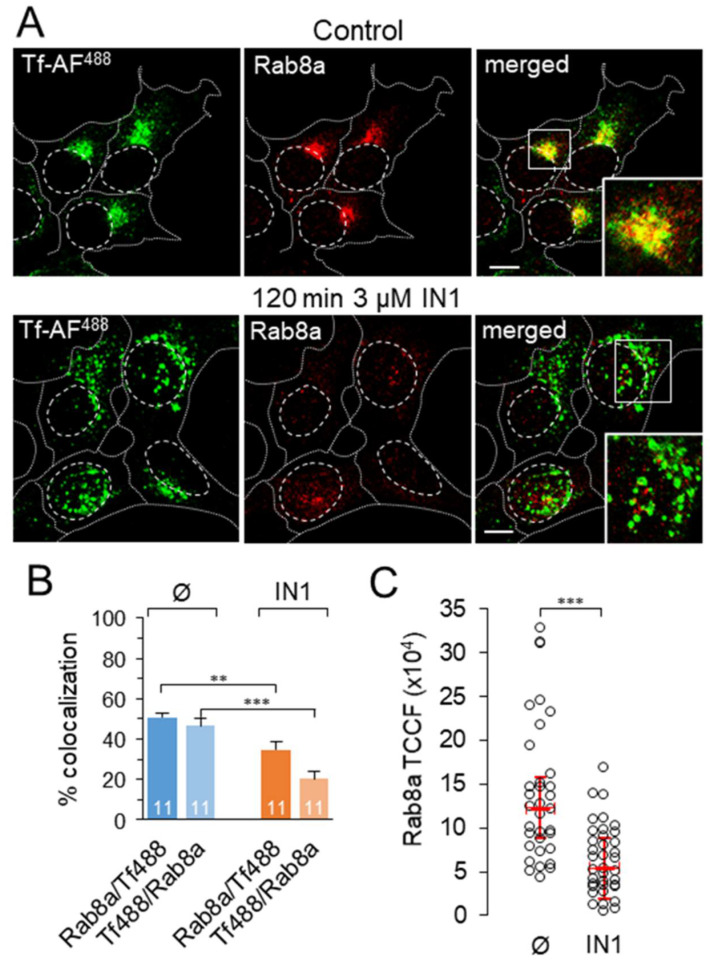
IN1 treatment depletes pericentriolar Rab8a-positive endosomes. (**A**) Representative images (*n* = 10–12) of internalized Tf-AF^488^ (45 min) and Rab8a in control and 120 min IN1-treated cells. (**B**) The 3D colocalization of TF-AF^488^ and Rab8a is based on the M1/M2 coefficients of pixel overlap measured across the Costes-algorithm thresholded *z*-stacks of confocal images. Data represent means ± SEM (number of cells indicated in bars); *** *p* ˂ 0.001; ** *p* ˂ 0.01 (two-tailed *t*-test). (**C**) Total corrected cell fluorescence (TCCF) of Rab8a in each cell was quantified using ImageJ on Otsu-thresholded confocal images. Data represent individual cells, and horizontal bars represent median value; *** *p* ˂ 0.001 (two-tailed *t*-test).

**Figure 7 cells-11-00962-f007:**
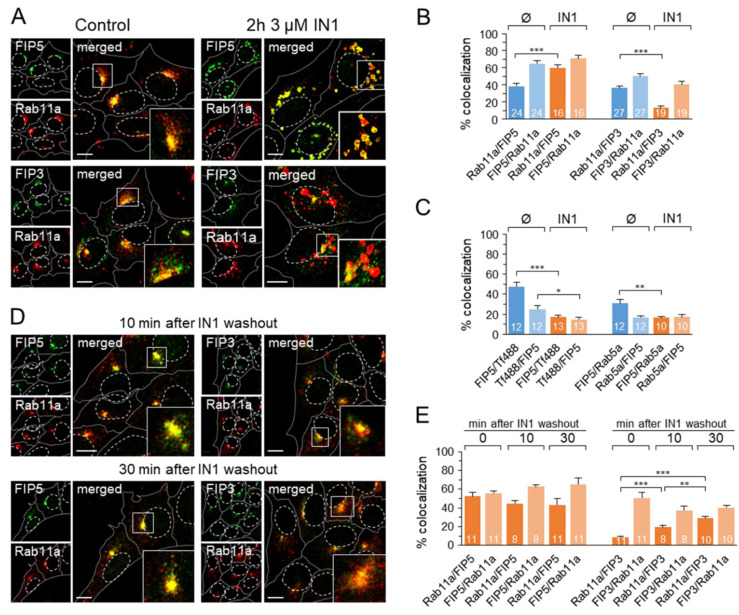
Rab11a endosomes of PI3P-depleted cells are positive for Rab11-FIP5 but not Rab11-FIP3. Representative images (**A**) and 3D colocalization analysis (**B**) of Rab11a and Rab11-FIP5 (FIP5) and Rab11-FIP3 (FIP3), in control (Ø) and 2 h IN1-treated cells (3 µM). Data represent mean ± SEM (number of cells indicated within bars) of M1/M2 coefficients of pixel overlap per cell, measured across the Costes-algorithm thresholded z-stacks of confocal images; *** *p* ˂ 0.001 (two-tailed *t*-test). (**C**) The 3D colocalization (M1/M2 coefficients of pixel overlap) of FIP5 with internalized Tf-AF^488^ (45 min internalization) and endogenous Rab5a in control (Ø) and 2 h IN1-treated cells (3 µM). Data represent mean ± SEM per cell (number of cells indicated in bars); *** *p* ˂ 0.001, ** *p* ˂ 0.01, * *p* ˂ 0.05 (two-tailed *t*-test). Representative images are presented in Appendix A. Representative images (**D**) and 3D colocalization analysis (**E**) of Rab11a and FIP5 or FIP3 in 2 h IN1-treated cells (3 µM) after 10 and 30 min of IN1 washout. Data represent mean ± SEM of M1/M2 coefficients of pixel overlap per cell (*n* = 8–11); *** *p* ˂ 0.001, ** *p* ˂ 0.01 (two-tailed *t*-test). The insets present zoomed area box. Cell borders are indicated by fine dotted lines and cell nuclei by fine dashed lines. Bars, 10 μm.

**Figure 8 cells-11-00962-f008:**
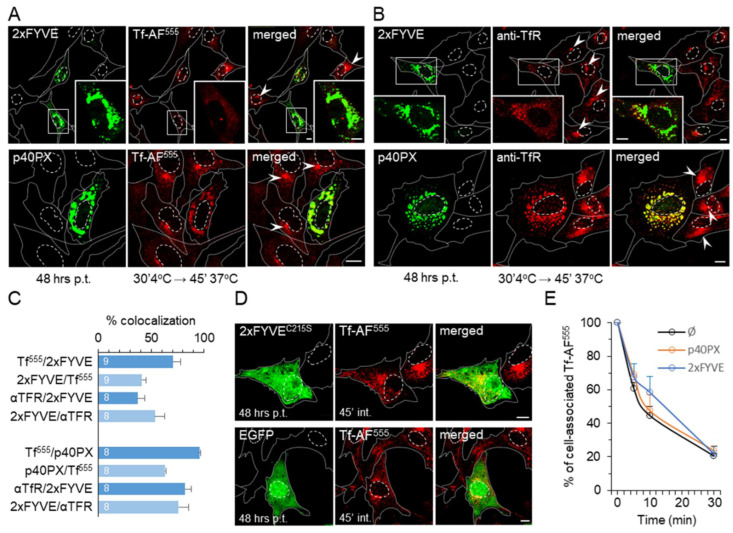
Prolonged expression of PI3P-binding domains alters TfR trafficking. Balb 3T3 cells were transfected with EGFP-2xFYVE MSCV or YFP-p40PX-MSCV and incubated 48 h post-transfection (p.t.) with Tf-AF^555^ (**A**) or anti-TfR mAb (**B**) for 30 min at 4 °C, followed by 45 min of internalization at 37 °C and confocal imaging. Arrowheads indicate juxtanuclear accumulation of internalized TfRs. (**C**) The 3D colocalization of 2xFYVE and p40PX in high expressing cells with internalized Tf-AF^555^ and mAb-TfRs based on the M1/M2 pixel overlap coefficients measured over the Costes-algorithm thresholded z-stacks of confocal images. Data are mean ± SEM per cell (number of cells indicated in bars). (**D**) Cells transfected with EGFP-2xFYVE^C215S^-MSCV or EGFP-MSCV were incubated with Tf-AF^555^ for 45 min and analyzed by confocal imaging. Cell borders are indicated by fine dotted lines and cell nuclei by fine dashed lines. Bars, 10 μm. (**E**) Kinetics of intracellular fluorescence loss by TfR recycling in untransfected (Ø) and EGFP-2xFYVE and YFP-p40PX transfected cells. The 48 h transfected cells were pulsed with hTf-AF^555^, chased in medium containing unlabeled hTf, and the intracellular fluorescence of transfected cells gated at the green channel was quantified by flow cytometry. Data represent mean ± SEM of five independent experiments.

**Figure 9 cells-11-00962-f009:**
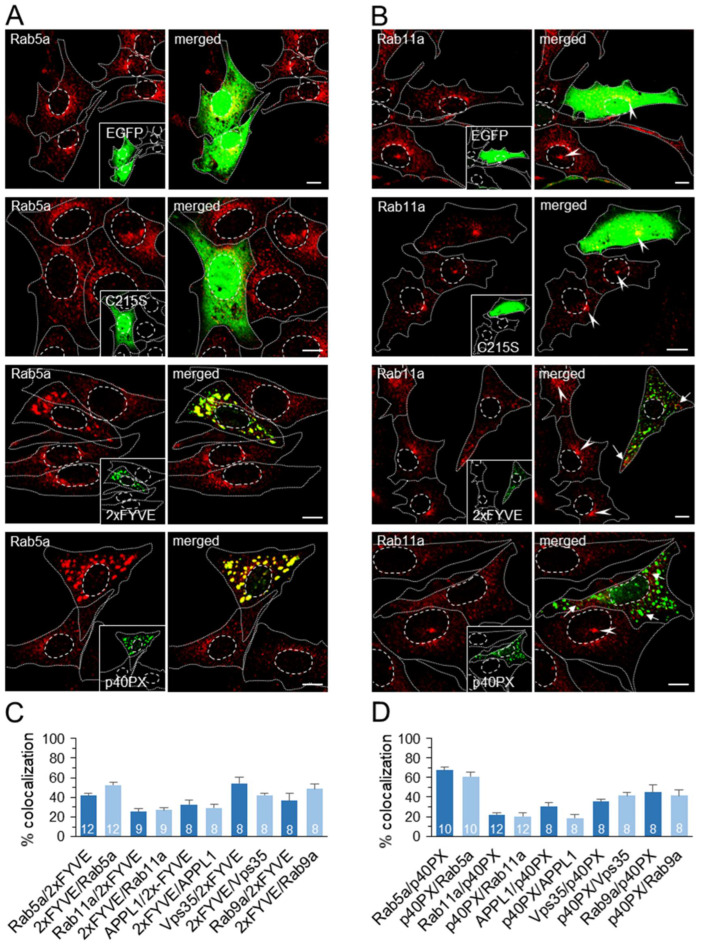
Colocalization of EGFP-2xFYVE and YFP-PX with Rab5a and Rab11a. (**A**,**B**) Balb 3T3 cells were transfected with either EGFP-MSCV, EGFP-2xFYVE^C215S^-MSCV (C215S), EGFP-2xFYVE-MSCV, or YFP-p40PX-MSCV for 48 h. Fixed and permeabilized cells were stained with rabbit Abs against Rab5a (**A**) and Rab11a (**B**) and secondary anti-rabbit AF^594^. Arrows point to peripheral Rab11a endosomes in 2xFYVE and p40PX expressing cells, and arrowheads point to the juxtanuclear and subnuclear ERCs. Cell borders are indicated by fine dotted lines and nuclei by fine dashed lines. Bars, 10 μm. (**C**,**D**) The 3D colocalization of Rab5a, Rab11a, APPL1, Vps35, and Rab9a with EGFP-2xFYVE (**C**) and YFP-p40PX (**D**) based on the M1/M2 coefficients of pixel overlap measured across the Costes-algorithm thresholded z-stacks of confocal images. Data are means ± SEM per cell (number of cells indicated in bars).

## Data Availability

The raw data supporting the conclusions of this article will be made available by the authors without undue reservation.

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
