# Peer review of "Early Endosomal Vps34-Derived Phosphatidylinositol-3-Phosphate Is Indispensable for the Biogenesis of the Endosomal Recycling Compartment"

_cells, 2022, doi:10.3390/cells11060962_

Round 1

Reviewer 1 Report

In this manuscript Marelić et al. use the Vps34 specific PI3 kinase inhibitor IN1 to study the role of PI3P in the biogenesis and recycling processes of early endosomes in mammalian cells. The authors start their analysis by characterizing the binding of PI3P specific probes in response to IN1 and extend this to endogenous PI3P binding proteins such as EEA1. As expected, the inhibition of Vps34 rapidly results in the loss of PI3P binding proteins from endosomal structures. The authors further use the transferrin receptor localization as a readout for early endosome functions and analyze the localization of multiple Rab proteins in the endosomes in response to PI3P kinase inhibition.

All carried out experiments focus on the localization and co-localization of multiple proteins of the early endosome. The experiments are all carried out carefully and the authors have to tried to do statistical analysis of the microscopy data as much as possible.

The manuscript contains some new information that could be potentially interesting to the trafficking field. However, in my opinion the manuscript has some serious flaws that are difficult to be addressed.

Major points

The manuscript appears to be very convoluted and it is difficult to distill the important information. Endosomal maturation and sorting is a highly dynamic process and it is important to study its functions with chemical PI3 kinase inhibitors rather than knockdowns or knockouts. The problem, in my opinion is, that these very dynamic processes are than analyzed in relatively static conditions at certain time points.

In my eyes the manuscript would tremendously benefit from simplification and cutting of a significant amount of data. The problem becomes obvious in the discussion where too many points are discussed and the actual message of manuscript is lost.

Another problem of the manuscript is described by the authors themselves stating that:” Nearly 80 cellular proteins use either FYVE or PX domains to bind PI3P”. Is it than possible to use co-localization studies of a sub-set of these PI3P binding proteins to study endosomal maturation and sorting?

Minor points:

As mentioned before, the authors quantify many of the observed phenotypes of their co-localization and localization studies. However, the gold standard for these analysis appear to be the so called superplots (J Cell Biol. 2020 Jun 1;219(6):e202001064. doi: 10.1083/jcb.202001064.). It would be great if, where appropriate, the data could be re-analyzed to give a better overview.

Reviewer 2 Report

Review of Manuscript Cells-1581976

The manuscript “Early endosomal Vps34-derived PI3P is indispensable for the biogenesis of the endosomal recycling compartment” from Marcelic and colleagues addresses the effects of acute short- and long-term inhibition of Vps34 and the concomitant depletion of endosomal PI3P on the biogenesis of the endosomal compartments and traffic through endosomal recycling routes. Interestingly, they find that although many PI3P-binding membrane-associated proteins are rapidly lost, cells partially adapt traffic through the endosomal compartment to a PI3P-independent configuration, which allows Tf recycling at normal rates, but strongly impairs lysosomal degradation and entry into Rab11a and Rab9-dependent recycling routes. They characterize this alternative pathway and find it to be to skip the pericentriolar Rab11a-positive ERC, instead remaining at an earlier, mostly Rab11a/Rab5a- and partially APPL1-positive state that is more dispersed in the cells, from which Tf and TfR recyle back to the PM. Finally, Marcelic and colleagues demonstrate that overexpression of artificial PI3P-binders causes a similar state, presumably by titrating away available PI3P binding sites in endosomal membranes.

The manuscript addresses an interesting and timely topic, and the data appear mostly sound. As the authors state and cite themselves, the effects of acute or chronic PI3P depletion have been addressed in prior publications, but the paper touches an interesting new and largely unexplored aspect. The ERC-independent recycling route has naturally not been characterized in every molecular detail here, but it demonstrates the versatility of endosomal sorting and will make an important contribution to the field. I have raised a few points below that I ask the authors to clarify (mostly changes in interpretation and presentation of the data) before I can recommend publication of the paper.

Major points:

Fig. 1C: Why is the total cellular fluorescence of EEA1 reduced if the total levels of EEA1 remain the same? There is a discrepancy here, which is probably derived from the image acquisition with automated background correction! This modus operandi can be quite problematic if an accurate quantification of imaging is required. Therefore, the authors should explain these drawbacks more prominently/explicitly. Also, there is considerable fluorescent spots in areas outside the confinement of the cells that look like aggregations of the secondary and/or primary antibody (or it may be further cells that are not labelled). This raises doubt about the specificity of the signals within the cells and the accuracy of background subtraction. Please clarify!

Fig. 2C: The quantitative analysis presented here seems questionable. E.g. the size of objects in IN1 treated cells is quantified to be smaller, but appears mostly increased to the eye in the presented image. It is doubtful if image J can accurately quantify individual spots of the compressed juxtanuclear objects in untreated cells. They are probably recognized as a single or few large objects, but this is probably not true and a limitation of the software or resolution. In the view of this reviewer the only conclusion that can be safely drawn is that the overall fluorescence intensity is not changed (2B) and that the objects disperse to outer areas of the cell (2D). Please consider to reanalyze or else remove Fig. 2C. This will not strongly affect the overall conclusions of Fig. 2.

Fig. 3: This figure was hard to assess because the quality of the jpg images provided by the authors is insufficient. E.g. the large subset of APPL-positive structures in untreated cells the authors mention in Fig. 1A is barely visible in the pdf. In a revised version please provide full quality images that have not undergone jpg compression for a final evaluation of the data.

In addition, some of the enlarged inlets seem not to correspond to the indicated regions marked by the boxes (e.g. in Fig. 3A and H; +IN1 images), or the regions of interest are not indicated at all (Fig. 3I without IN1).

Also, the labelling in Fig. 3E and J is probably incorrect. This is a ratio and not % of colocalization, or are all colocalization values below 1% of total signal? Moreover, rather than indicating the range of cell numbers used for quantification (i.e. 12-25), please indicate n for each bar separately to assess the power of the data. This applies also to all other figures with quantifications.

Fig. 4: It is unclear to the reviewer why the authors distinguish subnuclear from juxtanuclear positioning.Are these really functionally different compartments or is the difference merely in the eye of the observer? Both are probably equally close to the nucleus and also gamma-tubulin is more often than not observed below the nucleus. The analogy to the intermediate compartment of the Golgi seems a bit far-fetched. Please extend your thoughts and explain more explicitly why you think these are different states and why this could be functionally important.

Fig. 8: The pattern of Tf and TfR accumulation looks different in FYVE and PX-transfected cells, although the distribution of both overexpressed PI3P-binding domains looks rather similar. Sadly, in Fig. 8A,B  the authors do not show cells that were not transfected with the FYVE domain for comparison. These should be included here. Also, the Tf staining looks different between the FYVE-C215S transfected and the corresponding untransfected cells in Fig. 8D (Juxtanuclear vs. subnuclear), but the authors do not comment on that. Is it a representative image? Please clarify!

Minor points:

Fig. S1B: why are the cells expressing eGFP alone so much smaller? They appear hardly bigger than the nuclei of cells in other panels at the same magnification. Please clarify! Also, please enlarge the cells and present the image not so overexposed to accurately assess eGFP localization.

Fig. 1F – S3: according to Fig. S3 lanes 1, 4 and 5 are the relevant samples (labelled with control; 2h IN1; 12h IN1), but in Figure 1F lanes 3-5 from this blot appear to be shown. Is lane 1 or lane 3 the correct control of this experiment? It will not change the outcome of this experiment, but it needs to be correct.  Please clarify! If lane 1, 4 and 5 are the relevant ones, then please crop the blot 1F accordingly.

Fig. 3: The enlarged Rab5-positive vacuolized structures are reminiscent of structures that are frequently formed in cells expressing dominant active Rab5 mutants. They are found in numerous publications. It is not unlikely that Vps34 inhibition will result in a higher fraction of GTP-loaded Rab5 due to inefficient Rab conversion, since Vps34 is a Rab5 effector. This may also explain the inefficient recruitment of Rab7 to Rab5-positive structures. The authors could mention that analogy in favor of their hypothesis.

Fig. 9B: Now the authors no longer distinguish juxtanuclear and subnuclear Rab11a signal, but instead all signals are equally designated as juxtanuclear. Why?

Fig. 9: The authors might consider showing the Rab5 and Rab11-stained images enlarged, rather than the merged images, because it is mostly the changes in distribution of these markers that they want to illustrate. The merge rather hinders than helps, and a small panel would be sufficient to know which are the transfected cells.

Line 305 – pool instead of poll

Line 362: “recycling processes upstream of EE”. Are there any? Please cite!

Line 789/873: There is probably no conversion of CORVET into HOPS complexes (i.e. by subunit exchange); rather there is release of the former and recruitment of the latterm, as both are very stable and preformed complexes. Rephrase!

Line 889: It is unclear why the authors refer to Fig. 3 here. Do they mean their own Figure 3 or Fig. 3 of reference [77] or [78]? I did not find the discussed data in any of these figures. Please clarify!

Fig. 3: label “control” missing above panel F

Round 2

Reviewer 2 Report

all my concerns have been adequately addressed. I recommend publication of the work. Congratulations to the authors.